# Association between antecedent statin use and decreased mortality in hospitalized patients with COVID-19

Aakriti Gupta [1,2,3,6✉], Mahesh V. Madhavan [1,2,6], Timothy J. Poterucha[1], Ersilia M. DeFilippis[1], Jessica A. Hennessey[1], Bjorn Redfors[1,2,4], Christina Eckhardt[1], Behnood Bikdeli[1,2,3], Jonathan Platt[1], Ani Nalbandian [1], Pierre Elias [1], Matthew J. Cummings[1], Shayan N. Nouri[1], Matthew Lawlor[1], Lauren S. Ranard [1], Jianhua Li[1], Claudia Boyle[1], Raymond Givens[1], Daniel Brodie [1], Harlan M. Krumholz [3], Gregg W. Stone[2,5], Sanjum S. Sethi[1], Daniel Burkhoff[1,2], Nir Uriel[1], Allan Schwartz[1], Martin B. Leon[1,2], Ajay J. Kirtane[1,2], Elaine Y. Wan [1,7✉] & Sahil A. Parikh [1,2,7]

The coronavirus disease 2019 (COVID-19) can result in a hyperinflammatory state, leading to acute respiratory distress syndrome (ARDS), myocardial injury, and thrombotic complications, among other sequelae. Statins, which are known to have anti-inflammatory and antithrombotic properties, have been studied in the setting of other viral infections, but their benefit has not been assessed in COVID-19. This is a retrospective analysis of patients admitted with COVID-19 from February 1st through May 12th, 2020 with study period ending on June 11th, 2020. Antecedent statin use was assessed using medication information available in the electronic medical record. We constructed a multivariable logistic regression model to predict the propensity of receiving statins, adjusting for baseline sociodemographic and clinical characteristics, and outpatient medications. The primary endpoint includes in-hospital mortality within 30 days. A total of 2626 patients were admitted during the study period, of whom 951 (36.2%) were antecedent statin users. Among 1296 patients (648 statin users, 648 non-statin users) identified with 1:1 propensity-score matching, statin use is significantly associated with lower odds of the primary endpoint in the propensity-matched cohort (OR 0.47, 95% CI 0.36–0.62, p < 0.001). We conclude that antecedent statin use in patients hospitalized with COVID-19 is associated with lower inpatient mortality.

[1] NewYork-Presbyterian Hospital and the Columbia University Irving Medical Center, New York, NY, USA. [2] Cardiovascular Research Foundation, New York, NY, USA. [3] Yale Center for Outcomes Research and Evaluation, New Haven, CT, USA. [4] Department of Cardiology, Sahlgrenska University Hospital, Gothenburg, Sweden. [5] The Zena and Michael A. Wiener Cardiovascular Institute, Icahn School of Medicine at Mount Sinai, New York, NY, USA. [6] These authors contributed equally: Aakriti Gupta, Mahesh V. Madhavan. [7] These authors jointly supervised this work: Elaine Y. Wan, Sahil A. Parikh. ✉email: ag3786@cumc.columbia.edu; eyw2003@cumc.columbia.edu

Severe acute respiratory syndrome coronavirus 2 (SARS-CoV-2), the causal pathogen responsible for coronavirus disease 2019 (COVID-19), enters human cells by engaging angiotensin-converting enzyme 2 (ACE2)[1]. This interaction may lead to an imbalance of the renin-angiotensin-aldosterone system (RAAS), a potential mechanism contributing to this pathogen's virulence[2]. A combination of direct viral toxicity and indirect effects such as untempered thromboinflammation and dysregulation of the RAAS may underlie severe presentations of COVID-19, which can manifest with acute respiratory distress syndrome (ARDS), myocardial injury, and micro-and macro-thrombotic events[3]. As such, several anti-inflammatory, immunomodulatory, and antithrombotic therapies may hold promise for prevention or treatment of patients with COVID-19[4], and statins constitute one such class of medications.

Although statins have traditionally been administered to lower serum cholesterol, their pleiotropic effects, including anti-inflammatory and antithrombotic properties, make them an attractive class of drugs in the setting of COVID-19[5–8]. Moreover, through effects on lipid rafts in cellular membranes[9–12], statins may influence viral transmission and infectivity. By these putative mechanisms, statins appear to have the potential to mitigate the impact of myocardial injury and thrombotic events that are associated with severe COVID-19 presentations[13].

To date, a number of studies have evaluated the use of statins in the treatment of pneumonia and ARDS[5,14–20]. While primary results of randomized clinical trials evaluating statins in ARDS have not indicated a benefit[19,20], secondary analysis of 540 individuals from the HARP-2 (Hydroxymethylglutaryl-CoA Reductase Inhibition with Simvastatin in Acute Lung Injury to Reduce Pulmonary Dysfunction–2) trial demonstrated improved survival with statin treatment in patients with a hyperinflammatory phenotype[15].

In the current study, we compared patient baseline and presentation characteristics as well as clinical outcomes, including in-hospital mortality, stratified by antecedent statin use, in a cohort of patients admitted with COVID-19 to a quaternary academic medical center in New York City. We utilized propensity score matching and multivariable logistic regression to investigate the association of antecedent statin use with the primary endpoint of in-hospital mortality at 30 days and secondary endpoint of invasive mechanical ventilation at 30 days. Here we show that patients with antecedent statin use were generally older with more comorbidities, presented with lower levels of C-reactive protein at time of admission, and experienced lower inpatient mortality at 30 days in a propensity-matched cohort.

## Results

**Baseline patient characteristics**. Of 2626 patients included in the analysis, 951 (36.2%) were considered antecedent statin users (Table 1). On average, patients who were prescribed statins were older [median 70 (IQR 63–79) vs. 62 (49–76) years, p < 0.001] with no significant differences in sex (p = 0.06) or race/ethnicity (p = 0.12). Patients in the statin group were significantly more likely to have Medicare or Medicaid (63.0% vs. 53.6%) insurance, and less likely to be have commercial insurance (35.4% vs. 42.5%) (p < 0.001 for both). There was no significant difference in the New York City borough of residence in the two groups.

Furthermore, patients using statins were significantly more likely to have hypertension (74.0% vs. 43.3%), diabetes (55.8% vs. 26.1%), coronary artery disease (22.5% vs. 6.9%), heart failure (17.0% vs. 6.7%), and chronic kidney disease (22.0% vs. 9.6%) compared with patients not receiving statins (p < 0.001 for all). Similarly, patients receiving statins had higher rates of history of stroke/transient ischemic attack (13.9% vs. 5.6%) and atrial

arrhythmias (11.0% vs. 5.6%), p < 0.001 for both. There were no significant differences in liver disease between the two groups.

Patients on statins were significantly more likely to be prescribed ACEi (19.7% vs. 4.2%), angiotensin-receptor blockers (13.1% vs. 3.7%), P2Y12 inhibitors (11.9% vs. 1.1%), oral anticoagulants (20.3% vs. 12.3%), and beta-blockers (44.0% vs. 12.7%) as outpatients compared to those not taking statins (p < 0.001 for all). Of note, 77.0% of patients who were on antecedent statins and 8.6% of patients who were not on antecedent statins, received statins during hospitalization.

Furthermore, among the 850 patients for whom lipid levels were available, patients receiving statins had significantly lower mean low-density lipoprotein [77.9 (60.0–107.6) vs. 88.0 (67.0–117.0)] and total cholesterol levels [157.3 (127.7–191.0) vs. 164.9 (136.0–201.9)] compared with those who were not receiving them (p < 0.01 for all).

**Propensity-matched cohort characteristics**. Using 1:1 matching, a propensity-matched cohort of 1296 patients (648 patients on statins, 648 patients not on statins) was identified. No significant differences in demographics, comorbidities, or home medications remained in the propensity-matched cohort (Table 1). The distribution of the estimated propensity scores for receipt of statins among patients who did and did not receive outpatient statins is shown in Supplementary Fig. 1. In the matched analytic sample, the differences between pre-hospitalization variables were attenuated in the propensity-score-matched samples as compared with the unmatched samples (Supplementary Fig. 2).

At the time of initial presentation, patients receiving statins were less likely to present with tachypnea (22.1% vs. 28.7%, p < 0.01). There were no significant differences in the presence of fever, tachycardia, peripheral desaturation, or hypotension on initial assessment (Table 2).

In the propensity-matched cohort, patients on statins had significantly lower white blood cell count at presentation [7.6 (5.5–10.3) vs. 8.1 (5.8–11.6)] and lower CRP levels [100.0 (46.2–168.5) vs. 120.7 (61.2–194.9)] (p < 0.01 for both) (Table 2). There were no significant differences in high-sensitivity troponin T, D-dimer, ferritin, or ESR levels between groups.

**Clinical outcomes of propensity-matched cohort**. Differences in clinical outcomes in the propensity-matched sample are presented in Table 3. The primary endpoint occurred in 96 (14.8%) patients receiving statins compared to 172 (26.5%) not receiving statins, (OR 0.47, 95% CI 0.36–0.62, p < 0.001). The secondary endpoint occurred in 121 (18.6%) patients receiving statins compared to 142 (21.9%) not receiving statins, (OR 0.76, 95% CI 0.58–1.00). Patients with antecedent statin use had lower rates of in-hospital mortality at any time compared with individuals who were not on statins (20.8% vs. 33.7%, p < 0.001). There were no significant differences in invasive mechanical ventilation, vasopressor use, renal replacement therapy, or length of stay between the groups.

**Multivariable adjustment in overall cohort**. Statin use was significantly associated with a reduction in the primary endpoint (in-hospital mortality within 30 days) in the overall cohort in univariate (OR 0.69, 95% CI 0.56–0.85) and multivariable-adjusted analysis (OR 0.49, 95% CI 0.38–0.63) (Table 4). Other factors associated with increased odds of the primary endpoint included age, male sex, history of atrial arrhythmias, and diabetes (Fig. 1). Outpatient prescriptions of oral anticoagulants and P2Y12 inhibitor were also protective. These results were in agreement with sensitivity analyses performed within the dataset restricted to patients with hypertension, coronary artery disease

**Table 1 Baseline Characteristics in Unmatched and Propensity-Matched Cohorts of Patients Hospitalized with COVID-19.**

| Total N = 2626 | Unmatched | | | Matched | | |
|---|---|---|---|---|---|---|
| | Statin use 951 (36.2%) | No statin use 1675 (63.8%) | p value | Statin use (n = 648) | No statin use (n = 648) | p value |
| Demographics | | | | | | |
| Age, years, median (IQR) | 70 (63–79) | 62 (49–76) | <0.001 | 69 (61–77) | 71 (60–81) | 0.18 |
| Body mass index (kg/m$^2$) | 28.3 (24.7–32.8) | 27.9 (24.5–32.6) | 0.23 | 28.1 (24.7–32.4) | 27.1 (23.8–32.0) | 0.66 |
| Sex | | | 0.06 | | | 1.0 |
| Male | 519 (34.7%) | 978 (65.3%) | | 366 (56.5%) | 366 (56.5%) | |
| Female | 432 (38.2%) | 697 (61.7%) | | 282 (43.5%) | 282 (43.5%) | |
| Race/Ethnicity | | | 0.12 | | | 0.76 |
| Hispanic | 489 (51.4%) | 825 (49.2%) | | 327 (50.5) | 310 (47.8%) | |
| Non-Hispanic White | 88 (9.3%) | 149 (8.9%) | | 60 (9.3%) | 68 (10.5%) | |
| Non-Hispanic Black | 126 (13.2%) | 194 (11.6%) | | 87 (13.4%) | 87 (13.4%) | |
| Others/Missing | 248 (26.1%) | 507 (30.3%) | | 174 (26.9%) | 183 (28.2%) | |
| Location | | | 0.88 | | | 0.50 |
| Manhattan | 553 (58.1%) | 936 (55.9%) | | 371 (57.3%) | 381 (58.8%) | |
| Brooklyn | 24 (2.5%) | 46 (2.7%) | | 16 (2.5%) | 12 (1.9%) | |
| Queens | 29 (3.0%) | 47 (2.8%) | | 25 (3.9%) | 16 (2.5%) | |
| Bronx | 294 (3.1%) | 546 (3.3%) | | 197 (30.4%) | 209 (32.2%) | |
| Staten Island | 2 (0.2%) | 4 (0.2%) | | 2 (0.3%) | 1 (0.1%) | |
| Outside NYC | 49 (5.1%) | 96 (5.7%) | | 37 (5.7%) | 29 (4.5%) | |
| Insurance | | | <0.001 | | | 0.80 |
| Medicare/Medicaid | 599 (63.0%) | 896 (53.6%) | | 396 (61.1%) | 399 (61.6%) | |
| Commercial | 337 (35.4%) | 710 (42.5%) | | 241 (37.2%) | 235 (36.2%) | |
| Other/Uninsured | 15 (1.6%) | 65 (3.9%) | | 11 (1.7%) | 14 (2.1%) | |
| Comorbidities | | | | | | |
| Hypertension | 704 (74.0%) | 726 (43.3%) | <0.001 | 434 (67.0%) | 453 (69.9%) | 0.28 |
| Diabetes | 531 (55.8%) | 437 (26.1%) | <0.001 | 297 (45.8%) | 309 (47.7%) | 0.54 |
| Coronary artery disease | 214 (22.5%) | 115 (6.9%) | <0.001 | 96 (14.8%) | 91 (14.0%) | 0.75 |
| Heart failure | 162 (17.0%) | 113 (6.7%) | <0.001 | 91 (14.0%) | 78 (12.0%) | 0.32 |
| Chronic lung disease | 196 (20.6%) | 267 (15.9%) | <0.01 | 124 (19.1%) | 124 (19.1%) | 1.0 |
| Chronic kidney disease | 209 (22.0%) | 161 (9.6%) | <0.001 | 116 (17.9%) | 113 (17.4%) | 0.88 |
| Stroke/TIA | 132 (13.9%) | 93 (5.6%) | <0.001 | 68 (10.5%) | 67 (10.3%) | 1.0 |
| Atrial arrhythmias[a] | 105 (11.0%) | 118 (7.0%) | <0.001 | 64 (9.9%) | 61 (9.4%) | 0.85 |
| Liver disease | 31 (3.3%) | 53 (3.2%) | 0.98 | 22 (3.4%) | 21 (3.2%) | 1.0 |
| Home medications | | | | | | |
| ACE inhibitors | 187 (19.7%) | 70 (4.2%) | <0.001 | 76 (11.7%) | 63 (9.7%) | 0.28 |
| ARBs | 125 (13.1%) | 62 (3.7%) | <0.001 | 60 (9.3%) | 52 (8.0%) | 0.49 |
| P2Y12 inhibitors | 113 (11.9%) | 20 (1.1%) | <0.001 | 35 (5.4%) | 20 (3.1%) | 0.05 |
| Oral anticoagulants | 193 (20.3%) | 206 (12.3%) | <0.001 | 111 (17.1%) | 118 (18.2%) | 0.66 |
| Beta-blockers | 419 (44.0%) | 212 (12.7%) | <0.001 | 194 (29.9%) | 177 (27.3%) | 0.32 |
| Inpatient statins | 732 (77.0%) | 144 (8.6%) | <0.001 | 487 (75.1%) | 86 (13.3%) | <0.001 |
| Lipid profile (n = 850) | | | | | | |
| Total cholesterol | 157.3 (127.7–191.0) | 164.9 (136.0–201.9) | <0.01 | | | |
| Low-density lipoprotein | 77.9 (60.0–107.6) | 88.0 (67.0–117.0) | <0.01 | | | |
| High-density lipoprotein | 43.0 (34.0–54.4) | 42.0 (32.4–54.4) | 0.25 | | | |
| Triglycerides | 136.0 (98.0–187.8) | 136.0 (93.7–215.5) | 0.22 | | | |

Data are presented as N (%).

For age and body mass index, Student's two-sided t-test performed for hypothesis testing without adjustment for multiple comparisons. For all other variables, Pearson's two-sided chi-squared test used for hypothesis testing without adjustment for multiple comparisons.

ACE angiotensin-converting enzyme, ARB angiotensin-receptor blocker, IQR interquartile range, NYC New York City, TIA transient ischemic attack.

[a]Any atrial fibrillation, atrial flutter, and supraventricular tachycardia.

and stroke/transient ischemic attack, and in the dataset with modified definition of antecedent statin use (Supplementary Figs. 3 and 4), and if exposure variable was examined as inpatient statin use (Supplementary Fig. 5).

In addition, statin use tended to be associated with reduced odds of the secondary endpoint in the overall cohort in multivariable-adjusted analysis (OR 0.80, 95% CI 0.64–1.02), but was not statistically significant (Table 4).

## Discussion

The principal findings of this analysis of hospitalized patients with COVID-19 are (1) antecedent statin use was common in our cohort, as 36% of patients admitted to our institution were prescribed statins prior to their index admission; (2) patients receiving statins were older, with a higher burden of cardiovascular comorbidities, (3) patients receiving statins tended to present with lower levels of CRP, and (4) antecedent statin use was associated with significantly lower odds of patients experiencing the primary endpoint of in-hospital mortality in a propensity-matched analysis.

In addition to respiratory failure due to pneumonia and ARDS, COVID-19 is known to result in a number of extrapulmonary manifestations[21]. Posited mechanisms explaining the multiorgan dysfunction that can result from severe COVID-19 presentations include but are not limited to direct effects of SARS-CoV-2 infection as well as indirect effects of a dysregulated immune

**Table 2 Presenting vital signs and laboratory data in the propensity-matched cohort of patients hospitalized with COVID-19.**

| | Statins use (n = 648) | No statins use (n = 648) | p value |
|---|---|---|---|
| Presenting vital signs | | | |
| Temperature | 99.0 (98.2-100.1) | 98.8 (98.2-100.0) | 0.21 |
| Fever (temperature > 100.4 °F) | 139 (21.4%) | 129 (19.9%) | 0.54 |
| Respiratory rate | 18.0 (18.0-20.0) | 19.0 (18.0-22.0) | 0.19 |
| Tachypnea (RR > 21) | 143 (22.1%) | 186 (28.7%) | <0.01 |
| Oxygen saturation | 94.0 (90.0-97.0) | 94.0 (89.0-96.0) | 0.06 |
| Hypoxia (Oxygen saturation <92%) | 186 (28.7%) | 204 (31.5%) | 0.30 |
| Systolic blood pressure | 123.0 (109.0-139.0) | 124.0 (110.0 – 141.0) | 0.38 |
| Diastolic blood pressure | 72.0 (64.8-81.0) | 73.0 (65.0-81.0) | 0.08 |
| Hypotension (SBP < 90 mmHg) | 25 (3.9%) | 15 (2.3%) | 0.15 |
| Heart rate | 96.0 (86.0-109.3) | 98.0 (86.0-110.3) | 0.47 |
| Tachycardia (HR > 100) | 270 (41.7%) | 287 (44.3%) | 0.37 |
| Presenting laboratory values | | | |
| White blood cell count ($10^3$/μL) | 7.6 (5.5-10.3) | 8.1 (5.8-11.6) | <0.01 |
| Platelet count ($10^3$/μL) | 235.0 (159.0-329.0) | 224.0 (149.0-314.0) | 0.43 |
| Creatinine (mg/dL) | 1.8 (1.1-3.3) | 1.9 (1.1-3.8) | 0.19 |
| AST (U/L) | 81.0 (44.8-126.3) | 84.0 (45.5-142.5) | 0.32 |
| ALT (U/L) | 62.5 (33.3-95.8) | 60.5 (33.3-107.3) | 0.43 |
| Albumin (g/dL) | 3.6 (2.8-4.4) | 3.6 (2.7-4.4) | 0.93 |
| Lactate (mmol/L) | 2.8 (1.6-4.2) | 3.1 (1.8-4.7) | 0.35 |
| Hs-Troponin (ng/L) | 76.5 (34.3-164.8) | 88.0 (39.8-162.3) | 0.50 |
| D-dimer (μg/mL) | 2.1 (1.1-3.7) | 2.5 (1.3-5.0) | 0.37 |
| Ferritin (ng/mL) | 667.8 (335.1-1248.5) | 714.2 (368.2-1299.0) | 0.74 |
| ESR (mm/h) | 67.5 (37.8-97.3) | 66.5 (36.3-96.8) | 0.84 |
| CRP (mg/L) | 100.0 (46.2-168.5) | 120.7 (61.2-194.9) | <0.001 |

Data presented as N (%) or median (IQR).
For fever, tachypnea, hypoxia and hypotension, Pearson's two-sided chi-squared test used for hypothesis testing without adjustment for multiple comparisons.
For all other variables, Student's two-sided t test performed for hypothesis testing without adjustment for multiple comparisons.
*ALT* alanine transaminase, *AST* aspartate transaminase, *CRP* C-reactive protein, *ESR* erythrocyte sedimentation rate, *F* Fahrenheit, *HR* heart rate, *IQR* interquartile range, *IL-6* interleukin 6, *SBP* systolic blood pressure.

**Table 3 Clinical outcomes in the propensity-matched cohort of patients hospitalized with COVID-19.**

| | Statins use (n = 648) | No statins use (n = 648) | p value |
|---|---|---|---|
| Primary endpoint | 96 (14.8%) | 172 (26.5%) | <0.001 |
| Secondary endpoint | 121 (18.6%) | 142 (21.9%) | 0.17 |
| In-hospital mortality (any time) | 112 (17.2%) | 201 (31.0%) | <0.001 |
| Mechanical ventilation (any time) | 130 (20.1%) | 158 (24.4%) | 0.07 |
| Vasopressor use | 151 (23.3%) | 200 (30.9%) | <0.01 |
| CVVH | 37 (5.7%) | 45 (6.9%) | 0.42 |
| Length of hospital stay (days) | 7.0 (4.0-12.0) | 7.0 (3.0-14.0) | 0.27 |
| Days on ventilator | 13.5 (3.8-31.6) | 12.8 (2.6-34.7) | 0.77 |

Data presented as N (%) or median (IQR).
For length of hospital stay and days on ventilator, Student's two-sided t-test performed for hypothesis testing without adjustment for multiple comparisons. For all other variables, Pearson's two-sided chi-squared test used for hypothesis testing without adjustment for multiple comparisons.

response and hyperinflammatory state[21]. This is supported by clinical and laboratory markers of inflammation, as well as histopathologic and post-mortem data, which demonstrate extensive inflammation and endothelialitis as well as isolation of viral RNA in tissues from several organ systems in patients with COVID-19[21–25]. Therefore, identifying treatment strategies to prevent serious sequelae of this viral infection may have the potential to improve prognosis. The current analysis suggests that statins merit further evaluation in COVID-19 given their pleiotropic properties and potentially disease-modifying effects in the setting of this viral illness.

There are many potential explanations as to how statins may have contributed to lower 30-day in-hospital mortality in our cohort, despite high prevalence of cardiovascular comorbidities in patients with antecedent statin use. Statins, which target HMG-CoA (3-hydroxy-3-methylglutaryl coenzyme A) reductase, confer a significant mortality benefit in patients with atherosclerotic cardiovascular disease[5,26,27], who are overrepresented in hospitalized patients with COVID-19. In addition to hyperlipidemia and other cardiovascular risk factors, inflammation has been identified as a key modulator of atherogenesis and can contribute to adverse cardiovascular events[5,28,29]. The potential benefits from statins extend beyond cholesterol-lowering properties, as there is a robust literature supporting the anti-inflammatory properties of statins in the preclinical and clinical arenas, suggesting that these drugs can stabilize and restore endothelial function, and lower rates of circulating inflammatory biomarkers such as CRP[5,30]. In this regard, patients receiving statins

presented with significantly lower CRP levels in this cohort compared with those who were not on statins.

Plaque stabilization[31,32] and antithrombotic properties[33] are also favorable characteristics of this class of drugs. It has previously been demonstrated in several series of COVID-19 patients that pre-existing cardiovascular disease is associated with risk for

clinical decompensation and severe disease[13,34]. Therefore, it is conceivable that antecedent statin use may confer benefit by preventing myocardial injury and infarction as well as thrombotic events, both of which may have influenced mortality and endotracheal intubation rates.

Other mechanisms, which may explain the effects of statin use in patients with COVID-19, have also been suggested. Functional membrane microdomains or lipid rafts consist of cholesterol and sphingolipids[9–11], and viruses may gain entry to cells via receptors which are concentrated in these regions of the plasma membrane[12]. Thus, it has been theorized that statin-mediated reduction in cholesterol levels may sufficiently alter the makeup of these lipid rafts[12], potentially preventing or reducing likelihood for viral infection or replication, and hence disease severity. Though lipid levels were not available for our entire cohort, we did find lower levels of total cholesterol and low-density lipoprotein in statin users. In addition, a recent computational docking analysis was performed to assess the interaction between an important SARS-CoV-2 protease (Mpro)[35] and statins[36]. Interestingly, these authors found that several statins demonstrated stronger interactions with Mpro than some protease inhibitors, implicating a potential mechanism by which statins may be able to interfere with SARS-CoV-2 replication. Preclinical evidence suggests that statins (as with ACEi and ARBs) can

**Table 4 Associations between statin use with primary and second endpoints in propensity-matched cohort and multivariable-adjusted overall cohorts of patients hospitalized with COVID-19.**

|  | OR | 95% CI |
|---|---|---|
| Primary endpoint—In-hospital mortality within 30 days | | |
| PS-matched | 0.47 | 0.36–0.62 |
| Multivariable (PS-matched) | 0.48 | 0.35–0.65 |
| Multivariable (overall) | 0.49 | 0.38–0.63 |
| Secondary endpoint – Invasive mechanical ventilation within 30 days | | |
| PS-matched | 0.76 | 0.58–1.00 |
| Multivariable (PS-matched) | 0.89 | 0.68–1.2 |
| Multivariable (overall) | 0.80 | 0.64–1.02 |

*CI* confidence interval, *OR* odds ratio, *PS* propensity scoring.

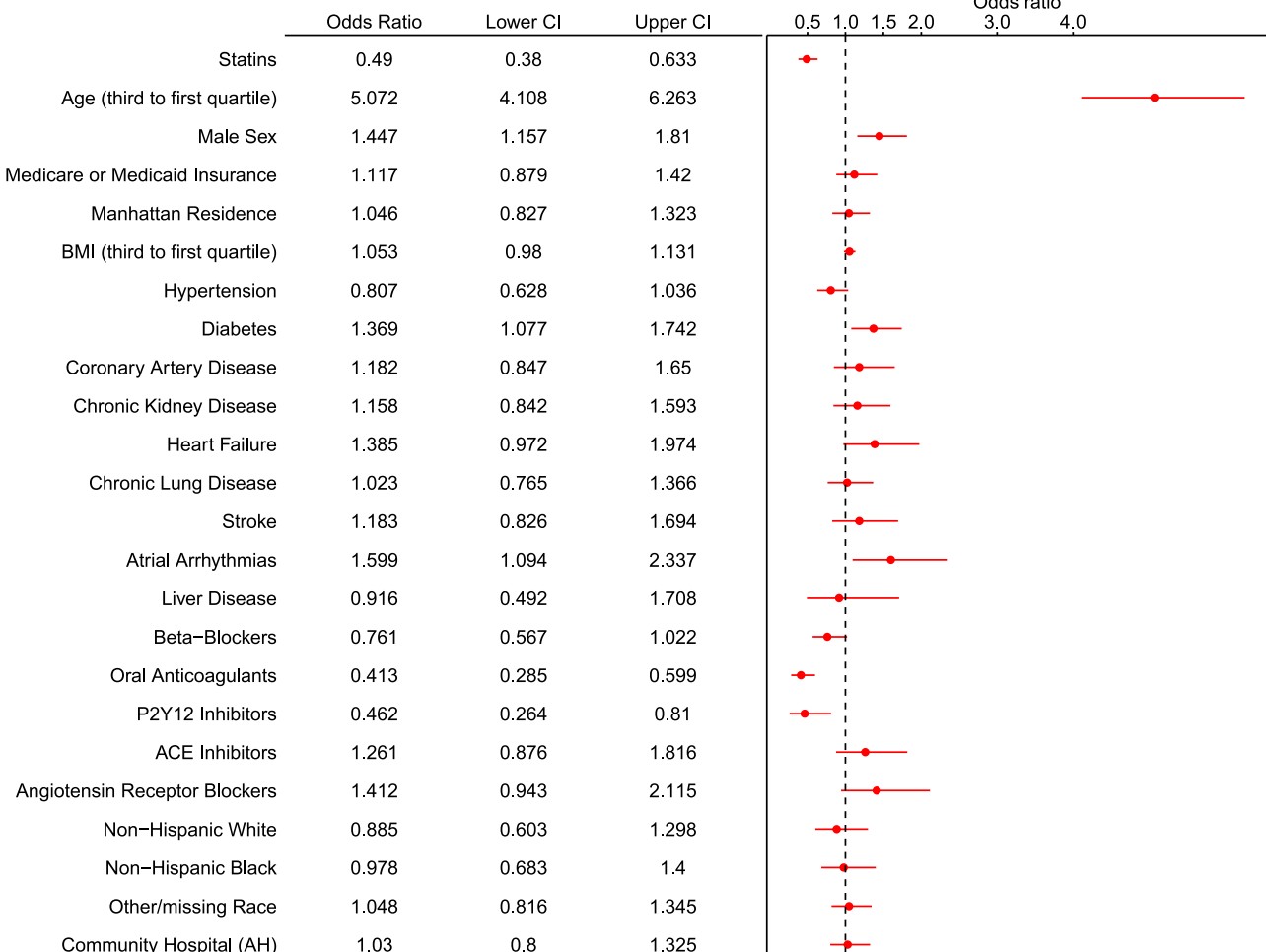

| | Odds Ratio | Lower CI | Upper CI |
|---|---|---|---|
| Statins | 0.49 | 0.38 | 0.633 |
| Age (third to first quartile) | 5.072 | 4.108 | 6.263 |
| Male Sex | 1.447 | 1.157 | 1.81 |
| Medicare or Medicaid Insurance | 1.117 | 0.879 | 1.42 |
| Manhattan Residence | 1.046 | 0.827 | 1.323 |
| BMI (third to first quartile) | 1.053 | 0.98 | 1.131 |
| Hypertension | 0.807 | 0.628 | 1.036 |
| Diabetes | 1.369 | 1.077 | 1.742 |
| Coronary Artery Disease | 1.182 | 0.847 | 1.65 |
| Chronic Kidney Disease | 1.158 | 0.842 | 1.593 |
| Heart Failure | 1.385 | 0.972 | 1.974 |
| Chronic Lung Disease | 1.023 | 0.765 | 1.366 |
| Stroke | 1.183 | 0.826 | 1.694 |
| Atrial Arrhythmias | 1.599 | 1.094 | 2.337 |
| Liver Disease | 0.916 | 0.492 | 1.708 |
| Beta−Blockers | 0.761 | 0.567 | 1.022 |
| Oral Anticoagulants | 0.413 | 0.285 | 0.599 |
| P2Y12 Inhibitors | 0.462 | 0.264 | 0.81 |
| ACE Inhibitors | 1.261 | 0.876 | 1.816 |
| Angiotensin Receptor Blockers | 1.412 | 0.943 | 2.115 |
| Non−Hispanic White | 0.885 | 0.603 | 1.298 |
| Non−Hispanic Black | 0.978 | 0.683 | 1.4 |
| Other/missing Race | 1.048 | 0.816 | 1.345 |
| Community Hospital (AH) | 1.03 | 0.8 | 1.325 |

**Fig. 1 Forest plot for in-hospital mortality within 30 days.** Forest plot demonstrating the odds ratio (OR) and 95% confidence interval (CI) for in-hospital mortality within 30 days with antecedent statin use (vs. no antecedent statin use) after multivariable logistic regression in the overall cohort. A number of other sociodemographic and baseline medication variables are also presented in this forest plot. ACE angiotensin-converting enzyme; AH Allen hospital. *N* = 2626 biologically independent patients.

contribute to increased ACE2 expression and epigenetic modification[37]. As ACE2 serves as the entry point for SARS-CoV-2 to human hosts, it remains to be completely understood how the modulation and modification of ACE2 levels may impact viral replication and infectivity.

In addition to the mechanisms above, studies prior to the current COVID-19 era evaluated the use of statins to prevent progression of ARDS and limit severity of illness[5,15,19,38,39], and evidence in this space is incomplete. While primary results of the randomized SAILS (Statin for Acutely Injured Lungs From Sepsis) and HARP-2 trials did not demonstrate statins were beneficial in ARDS[19,20], a secondary analysis of HARP-2 suggested improved survival with statin use in a hyperinflammatory phenotype[15]. Subphenotype analysis of SAILS did not replicate these results[40]. It is important to note that these trials administered different drugs and were characterized by different inclusion criteria[41]. Whether specific statins confer greater benefit due to higher bioavailability in lung tissue or more profound pleiotropic effects remains to be seen. However, the benefit noted from statins in our cohort may share mechanisms with some of the prior positive statin studies in the ARDS literature.

The limited evidence available regarding statins in the COVID-19 literature confirms the findings presented in the present manuscript. In a study which evaluated the prevalence and impact of myocardial injury in 2736 hospitalized patients in New York City, 36% of patients received statins prior to admission[42]. Though not the focus of this manuscript, statin use was associated with significantly lower rates of in-hospital mortality by multivariable analysis (OR 0.57, 95% CI 0.47–0.69)[42]. In addition, a separate study of 154 elderly individuals suggested that statin use prior to admission was associated with less severe symptoms, but they did not assess in-hospital mortality[43]. More recently, an analysis from the Wuhan, China demonstrated significantly lower 28-day mortality in patients who received inpatient statins compared with non-statin users (adjusted hazard ratio 0.58, 95% CI 0.43–0.80)[44]. In this study, however, <10% of hospitalized patients received statins. A meta-analysis of 8990 patients from four retrospective studies (including the study by Zhang et al.[44]) revealed that COVID-19 patients who were statin users experienced significantly lower hazard for death or severe disease compared with non-statin users (hazard ratio 0.70, 95% CI 0.53–0.94)[45]. As the majority of these studies focused on patients from China, they may not be representative of the patient characteristics and burden of cardiovascular comorbidities in Western populations. Most recently, a separate meta-analysis focused exclusively on European and North American patient populations, and only one of the seven studies included overlapped with the previously mentioned analysis by Kow et al.[45,46]. Statin use was associated with significantly lower rates of progression to severe COVID-19 illness or death (OR 0.59, 95% CI 0.35–0.99)[46]. Notably, studies included in both of these meta-analyses varied significantly in terms of patient populations, adjunctive therapies administered, timing of administration (inpatient vs. outpatient) as well as drug and dosing of statin regimens. Importantly, as in the study by Zhang et al.[44], in-hospital statin use in an observational setting may be subject to immortal time bias. With these studies as well as the findings of the present analysis in mind, the results of ongoing randomized clinical trials and registries will be crucial (Clinicaltrials.gov Identifiers: NCT04407273, NCT04390074, NCT04348695, NCT04426084, NCT04333407, NCT04380402, NCT04486508)[47].

Our study has important limitations. We performed propensity matched analysis and multivariable adjustment to minimize the likelihood for confounding. As a retrospective analysis of electronic medical record data, however, there remains the potential for unmeasured confounders. In addition, we also performed a number of sensitivity analyses and findings remained consistent. In addition, medication reporting and reconciliation in the electronic medical record, especially in the setting of the ongoing pandemic, may have been subject to errors on the part of individual clinicians. While the primary endpoint of in-hospital mortality was significantly lower in antecedent statin users, it remains to be seen whether patients who survived (possibly in-part due to prior statin therapy) may experience long-term morbidity and sequelae of COVID-19 infection, and further analyses are needed in this regard. Moreover, it was not possible to verify duration of statin therapy or patient adherence with statin therapy. However, patients in the antecedent statin group had better lipid profiles, suggestive of medication effect. In addition, errors with data entry are unlikely to affect the primary endpoint, inpatient mortality. To increase the capacity for critical care interventions, multiple temporary intensive care units (ICU) were created in previously non-ICU patient care areas at our institution. Thus, assessment for need for ICU level-of-care or outcomes in ICU patients was not possible. We did, however, assess need for invasive mechanical ventilation as part of the key secondary endpoint, a therapy which was almost exclusively used in ICU settings. Further, patients who are receiving statins may reflect overall better outpatient care, and possibly a low-risk cohort. However, patients on statins were much older, and had a more severe burden of comorbidities in our study. Moreover, we examined proxy variables for socioeconomic status, including NYC borough of residence and medical insurance, and these were not significantly different between the two groups. In addition, missingness for disease and drug variables cannot be quantified, as there were no codes to indicate that data were missing. It was assumed that a characteristic was not present if the patient's record did not include information on it, such as hypertension or the use of statins. Lastly, a small proportion of patients (2.5%) included in this analysis remained hospitalized at the end of study period, and so reporting on in-hospital outcomes in such patients remains incomplete at this time.

In this large analysis from a quaternary academic medical institution in an epicenter of the COVID-19 pandemic, we demonstrated that antecedent statin use was associated with significantly lower rates of in-hospital mortality within 30 days. These results indicate the important need for randomized controlled trials evaluating the benefits of statin therapy in patients affected by COVID-19.

## Methods

**Patient population and data elements.** For this retrospective study, we utilized data from the Columbia University Irving Medical Center (CUIMC) and Allen Hospital sites of the NewYork-Presbyterian Hospital (NYPH). Adult patients (≥18 years of age) who were hospitalized between February 1 through May 12, 2020, and tested positive for SARS-CoV-2 reverse transcriptase-polymerase chain reaction testing of nasopharyngeal or oropharyngeal specimens were included in the present analysis. All testing was conducted either by NYPH laboratories or the New York State Department of Health (in the period of time prior to when internal testing capabilities were available). Patients who were admitted for less than 24 h were excluded from this analysis. The mortality rate was not significantly different between statin users and non-statin users who were discharged within 24 h and excluded from our analysis (Supplementary Table 1). The study period ended on June 11th, 2020, allowing for a follow-up period of at least 30 days in all patients. The CUIMC Institutional Review Board approved this study and waived the requirement for obtaining informed consent. Deidentified data will be made available on request to the corresponding authors.

Patient data were identified in the electronic medical record by using the institution's clinical data warehouse, which includes outpatient and inpatient information on individuals who receive care at our institution. No manual chart abstraction was performed. Follow-up for each patient continued until patients were discharged, died in-hospital, or the end of the study period was reached. Data analysis was limited to the index hospitalization in the event of readmissions. Baseline information including age, sex, race and ethnicity, insurance, New York City borough of residence, body mass index (BMI), comorbidities, and outpatient medications were recorded. Clinical comorbidities, including hypertension,

diabetes, coronary artery disease, heart failure, stroke or transient ischemic attack, atrial arrhythmias (atrial fibrillation, atrial flutter and supraventricular tachycardia), chronic lung disease, chronic kidney disease, and chronic liver disease were identified using ICD-10 medical billing codes (Supplementary Table 2). Outpatient medications, including statins, angiotensin-converting enzyme inhibitors (ACEi), angiotensin receptor blockers (ARB), beta-blockers, oral anticoagulants, and P2Y12 inhibitors were extracted from medication reconciliation fields in the electronic medical record, which are entries of current prescriptions that are updated at the time of hospital admission.

We included features of the clinical presentation, including vital signs at presentation (i.e., temperature, heart rate, blood pressure, respiratory rate, and peripheral oxygen saturation). Several laboratory parameters at presentation were also collected from the electronic medical record, including white blood cell count, platelet count, creatinine, hepatic panel tests including aspartate aminotransferase (AST), alanine aminotransferase (ALT) and albumin, lactate, high-sensitivity troponin T, D-dimer, ferritin, erythrocyte sedimentation rate (ESR), and C-reactive protein (CRP). Given that statins may act by lowering lipid levels, we collected lipid values for patients from inpatient and outpatient records at any dates after January 1, 2018, and averaged them for each patient over the study period. As some patients did not have all laboratory studies of interest collected as part of clinical care, data are presented for only the patients in whom these were available. Details for missing laboratory values are provided in Supplementary Table 3.

**Study exposure**. The exposure in this study was antecedent statin use. Antecedent statin use was defined as record of current prescription of statins as a home medication in the electronic medical record. Home medications are typically reconciled with patients or their families or pharmacies at the time of admission.

**Study outcomes**. The principal outcome was in-hospital mortality within 30 days of admission. The secondary outcome was invasive mechanical ventilation within 30 days of admission. Other outcomes included in-hospital mortality at any time and invasive mechanical ventilation at any time, as some patients had a length of stay longer than 30 days. We also examined hospital length of stay (days), duration of invasive mechanical ventilation (days), renal replacement therapy with continuous veno-venous hemofiltration, and use of vasopressors.

**Statistical analysis**. We examined differences in sociodemographic, baseline clinical characteristics, and outpatient medications by antecedent statin use. Summary statistics are presented as numbers and percentages for categorical variables and medians and interquartile ranges for continuous variables. Differences between groups were examined using the two-sided independent $t$-test and chi-squared test, as appropriate.

To address confounding by indication, we constructed a multivariable logistic regression model to predict the propensity of antecedent statin administration, adjusted for the following variables: age, sex, first BMI assessment, race and ethnicity, insurance, New York City borough of residence, history of hypertension, diabetes, coronary artery disease, heart failure, stroke or transient ischemic attack, atrial arrhythmias, chronic lung disease, chronic kidney disease, and liver disease; outpatient use of beta-blockers, ACEi, ARBs, oral anticoagulants, and P2Y12 receptor inhibitors. Propensity-score matching was implemented with the use of a nearest-neighbor strategy with specification of caliper width equal to 0.1 of the standard deviation of the logit of the propensity score. Descriptive analyses were performed for all baseline variables in the propensity-matched cohort.

For the primary and secondary endpoints, we performed logistic regression on the propensity-matched cohort with the control group as reference. In addition, to examine whether the effect estimate remained consistent in the overall cohort, we performed logistic regression with multivariable adjustment on the overall cohort. We adjusted the multivariable models for variables that have been previously studied in association with mortality in COVID-19 including baseline sociodemographic and clinical characteristics and outpatient medications.[48,49]

**Sensitivity analyses**. We performed sensitivity analyses by defining any recent statin use as either antecedent statin or inpatient statin use. We also examined the data by defining statin use as inpatient statin use. Using these modified definitions, we evaluated the association of any recent statin use with the primary endpoint using multivariable logistic regression. We also performed subgroup analyses to assess the association of antecedent statin use with primary endpoint in a subset of patients with history of hypertension, coronary artery disease and stroke, conditions for which statins are usually prescribed.

**Missing data**. BMI and insurance information were missing in 19% and 15% of the patients, respectively, and multiple imputation with predictive mean matching was utilized to adjust the models for BMI and insurance. We imputed 100 datasets, fitted the logistic regression models for the primary and secondary endpoints for each imputed dataset, estimated the odds ratios on each imputed dataset, and then averaged the one hundred estimated values to obtain the pooled estimates. Model estimates and standard errors were calculated with Rubin's rules[11]. Race and ethnicity were missing in 30% of the patients and were classified as 'others/missing' while adjusting in the models. Lipid levels were available for only 32% of the

cohort. As such, we have presented them only at baseline. The remaining variables were missing in fewer than 5% of the study cohort.

$P$ values <0.05 were considered significant for the analysis. We did not adjust for multiple comparisons as this was an exploratory analysis. All analyses were performed using version 3.5.1 of the R programming language (R Project for Statistical Computing; R Foundation, mice, MatchIt, cobalt, rms packages).

**Reporting summary**. Further information on research design is available in the Nature Research Reporting Summary linked to this article.

## Data availability

Deidentified data will be made available upon request to the corresponding authors. These cannot be made publicly available given institutional restrictions.

## Code availability

The analysis code utilized for this study will be made available upon request to the corresponding authors.

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

## Author contributions

A.G.—designed the study, performed data analysis and drafted the manuscript. M.V.M.—designed the study and drafted the manuscript. T.J.P.—drafted the manuscript and helped with critical revision of manuscript for intellectual content. E.M.D.—drafted the manuscript and helped with critical revision of manuscript for intellectual content. J.A.H.—drafted the manuscript and helped with critical revision of manuscript for intellectual content. B.R.—designed the study and helped with critical revision of manuscript for intellectual content. C.E.—critical revision of manuscript. B.B.—designed the study and helped with critical revision of manuscript for intellectual content. J.P.—helped with data analysis and review of methods section of manuscript. A.N.—critical revision of manuscript for intellectual content. P.E.—critical revision of manuscript for intellectual content. M.J.C.—critical revision of manuscript for intellectual content. S.N.N.—critical revision of manuscript for intellectual content. M.L.—critical revision of manuscript for intellectual content. L.S.R.—critical revision of manuscript for intellectual content. J.L.—data acquisition and analysis. C.B. data acquisition. R.G.—helped with study design. D.Brodie, —critical revision of manuscript for intellectual content. H.M.K.—study design and critical revision of manuscript for intellectual content. G.W.S.—study design and critical revision of manuscript for intellectual content. S.S.S.—critical revision of manuscript for intellectual content. D.Burkhoff—critical revision of manuscript for intellectual content. N.U.—critical revision of manuscript for intellectual content. A.S.—critical revision of manuscript for intellectual content. M.B.L.—critical revision of manuscript for intellectual content. A.J.K.—study design and critical revision of manuscript for intellectual content. E.Y.W.—provided funding for publication, helped with study design and critical revision of manuscript for intellectual content. S.A.P.—study design and critical revision of manuscript for intellectual content.

## Competing interests

A.G. received payment from the Arnold & Porter Law Firm for work related to the Sanofi clopidogrel litigation and from the Ben C. Martin Law Firm for work related to an inferior vena cava filter litigation; received consulting fees from Edward Lifesciences; and holds equity in the healthcare telecardiology startup Heartbeat Health. M.V.M. has received support from an institutional grant by the National Institutes of Health/National Heart, Lung, and Blood Institute to Columbia University Irving Medical Center (T32 HL007854). B.B. reports that he is a consulting expert, on behalf of the plaintiff, for litigation related to a specific type of IVC filter. M.J.C. reports being a co-investigator for clinical trials evaluating the efficacy and safety of Remdesivir (Gilead Sciences) and convalescent plasma (Amazon) in hospitalized patients with COVID-19. Support for this work, which is unrelated to the current study, is paid to Columbia University. D.B. receives research support from ALung Technologies, he was previously on their medical advisory board. He has been on the medical advisory boards for Baxter, BREETHE, Xenios and Hemovent. G.W.S. reports speaker or other honoraria from Cook, Terumo, QOOL Therapeutics and Orchestra Biomed; Consultant to Valfix, TherOx, Vascular Dynamics, Robocath, HeartFlow, Gore, Ablative Solutions, Miracor, Neovasc, V-Wave, Abiomed, Ancora, MAIA Pharmaceuticals, Vectorious, Reva, Matrizyme, Cardiomech; equity/options from Ancora, Qool Therapeutics, Cagent, Applied Therapeutics, Biostar family of funds, SpectraWave, Orchestra Biomed, Aria, Cardiac Success, MedFocus family of funds, and Valfix. T.J.P. owns stock in Abbott Laboratories, AbbVie, Inc, Baxter International, and Edwards Lifesciences. H.M.K. works under contract with the Centers for Medicare & Medicaid Services to support quality measurement programs; was a recipient of a research grant, through Yale, from Medtronic and the U.S. Food and Drug Administration to develop methods for post-market surveillance of medical devices; was a recipient of a research grant with Medtronic and is the recipient of a research grant from Johnson & Johnson, through Yale University, to support clinical trial data sharing; was a recipient of a research agreement, through Yale University, from the Shenzhen Center for Health Information for work to advance intelligent disease prevention and health promotion; collaborates with the National Center for Cardiovascular Diseases in Beijing; receives payment from the Arnold & Porter Law Firm for work related to the Sanofi clopidogrel litigation, from the Ben C. Martin Law Firm for work related to the Cook Celect IVC filter litigation, and from the Siegfried and Jensen Law Firm for work related to Vioxx litigation; chairs a Cardiac Scientific Advisory Board for UnitedHealth; was a participant/participant representative of the IBM Watson Health Life Sciences Board; is a member of the Advisory Board for Element Science, the Advisory Board for Facebook, and the Physician Advisory Board for Aetna; and is the co-founder of HugoHealth, a personal health information platform, and co-founder of Refactor Health, an enterprise healthcare AI-augmented data enterprise. Other authors report no disclosures. A.J.K. reports institutional funding to Columbia University and/or Cardiovascular Research Foundation from Medtronic, Boston Scientific, Abbott Vascular, Abiomed, CSI, Philips, ReCor Medical.
