## [Peer Review File · Nature Communications]

Reviewers' Comments:

Reviewer #1:

Remarks to the Author:

Using data from Electronic Health Records, this study did a retrospective investigation to evaluate the effect of statins use on lowering the in-hospital mortality rate of COVID-19. Despite intensive discussions on this topic recently, real-world evidence demonstrating the effect of statins on COVID-19 is still lacking. This paper is of clinical importance, but can be improved by addressing the following comments.

1. There are a few recently published studies investigating the effect of statins use on COVID-19. Although the population might not be the same as this study, I think the authors should compare the results and provide some discussions. For example, Kow CS, Hasan SS. Meta-analysis of Effectiveness of Statins in Patients with Severe COVID-19. *American Journal of Cardiology*. 2020 Aug 12; Rodriguez-Nava, G., Trelles-Garcia, D.P., Yanez-Bello, M.A., Chung, C.W., Trelles-Garcia, V.P. and Friedman, H.J., 2020. Atorvastatin associated with decreased hazard for death in COVID-19 patients admitted to an ICU: a retrospective cohort study. *Critical Care*, 24(1), pp.1-2.
2. The secondary outcome is "a composite of in-hospital mortality or invasive mechanical ventilation within 30-days of admission". So, my understanding is that the authors combined in-hospital mortality and invasive mechanical ventilation, which does not make sense here. Why not just use invasive mechanical ventilation within 30-days as the secondary outcome? From Table 3, it is clear that statins use is only associated with mortality but not the invasive mechanical ventilation, if we subtract the primary endpoints from the secondary endpoints. The significant association of the second endpoint comes from the association between the primary endpoint and statins use.
3. In Table 3, the p-value of oxygen saturation is a bit counterintuitive. The two groups have the same mean for oxygen saturation (both 94.0), why the p-value is so small? Compared with oxygen saturation, temperature has bigger mean difference and smaller confidence interval (which implies smaller variance), why the p-value is larger than oxygen saturation?
4. The data used in this analysis also provide opportunities for mediation analysis to understand how statins impact the mortality of COVID-19 patients. Since the lipid levels were observed, it would be interesting to investigate whether statins reduce the risk of mortality through lowering lipid level or other potential biological pathways.

Reviewer #2:

Remarks to the Author:

Please see attached file.

Gupta et al. performed a retrospective analysis of patients admitted with COVID-19 from February 1 to May 12 2020, to determine whether antecedent statin use was associated with lower 30-day in-hospital mortality in patients with confirmed COVID-19 infection. They report results on a total of 2,626 patients and a sub-cohort of 1,296 propensity-matched patients on 30-day hospital mortality (primary endpoint) and multiple secondary outcomes including a combination of hospital-mortality and various organ support measures. Based on their analysis, the authors demonstrate a statistically significant association of antecedent statin use with lower 30-day hospital mortality and advocate future prospective randomized controlled trials to confirm these findings.

This is an interesting and hypothesis generating study, particularly given the scarcity of treatment options and preventive therapeutic strategies for COVID-19. Statins are readily available and cheap, and these findings are encouraging and warrant further confirmation.

1. Statins may affect various aspects of COVID-19 – reducing risk of acquiring COVID-19 infection, reducing severity and need for hospitalization, and once hospitalized, may reduce risk of poor outcomes. Although the authors identified statin use at home, it is not possible to draw inferences about statins reducing risk of acquiring COVID-19 infection or severity to prevent hospitalization since only hospitalized patients were assessed.

2. There are several concerns with regards to the analyses to determine the association between statins and in-hospital mortality.

a) The focus should be on patients who received statins during the hospitalization rather than all patients who received statins at home.

b) Statin users were older and had more chronic diseases. They may have presented earlier after symptom onset or more likely to be admitted – hence resulting in lead time bias. Non-statin users had higher point estimates for several important biomarkers (table 2 suggests they may be sicker at presentation), even after propensity matching, which corroborates this concern. Perhaps propensity matching of patients based on pre-admission characteristics and vital signs on presentation (Table 2). While this approach may include some variables that may mediate the beneficial effects of statins, it would address the issue of lead time bias.

c) Was the median time between COVID-19 testing (presumed symptom onset) and hospital admission similar in both groups. Were all COVID-19 tests performed in the emergency department or were patients referred to the ED after community-based testing?

d) No information is provided on the quality of patient care other than organ support measures during hospitalization. Did statin and non-statin users receive similar supportive care and COVID-19 treatments? Therapeutic anticoagulation? Steroids? Fluid management? Or were statin-users perhaps 'more aggressively' treated early on based on their perceived higher risk?

e) Quality of outpatient and inpatient care may not only vary within hospitals, but also across hospitals. The authors should consider accounting for hospital or 'center' effect.

3. Other concerns.

a) The time on ventilator and 30-day hospital mortality could be combined as an alternative secondary outcome of 'ventilator-free days.'

b) Discussion could be shortened.

c) I didn't see description of statin type or dose. Was a dose-response relationship observed?

Reviewer #3:

Remarks to the Author:

This paper seeks to characterize the effect of statin use on COVID-19 health outcomes using data from a retrospective cohort assembled from the electronic health records of an academic center. Propensity score matching methods were used to account for the observational nature of this study.

The topic of this study is certainly salient and timely. There has been mounting evidence that statins may indeed be protective of various adverse outcomes in the context of COVID-19. This study could add one more pixel to the emerging picture, albeit one that may only be suggestive since it is based on observational data (and indications for statins are themselves strong risk factors for severe COVID-19). Ongoing clinical trials will hopefully provide a more definitive assessment and clinical guidance in due time.

There are several comments/questions that I noted while reading this manuscript.

1. The definition of the exposure is a bit perplexing to me for several reasons.

a) Are patients deemed to be antecedent statin users if they have ever been recorded to use statins, or does the definition limit how far back in time such a record must have been made? It would help to further clarify the definition of an antecedent statin user.

b) It is unclear to me what potential intervention(s) involving statins the authors hypothesize as being possibly useful. For example, are they imagining an eventual recommendation for all at-risk individuals to be prescribed/administered statins as prophylaxis, at first symptoms of infection, or at hospital admission? This is important since it determines the relevant definition of the exposure, and statistical analyses should reflect this definition.

c) In its current definition, the exposure appears to (possibly) occur over a (patient-specific) length of time prior to baseline rather than at baseline. In the propensity score model, variables that are in the causal pathway between statin use and COVID-19 outcomes should not be included. However, antecedent statin use not only precedes (and thus may affect) baseline patient characteristics, but also certain pre-baseline measurements obtained from the medical records. It would seem important to provide the hypothesized causal diagram underlying the mechanisms under study, and to comment on the adequateness of adjusting for each variable currently included in the model.

2. It is not clear to me what the source of medical records used to gather exposure and other patient data is. I presume records used are from the academic center in question. However, since it does not appear that this center is part of a closed, integrated care system (of the Kaiser-Permanente type), I wonder how complete such records are, and what population those patients for whom records are indeed available are representative of. Are most patients in this cohort seen in primary care at this same academic center? Additionally, for information obtained at admission, would there not be the risk of informative missingness, since patients admitted in more severe condition may be unable to provide much information, if any at all, particularly given restrictions on access to ERs and ICUs in the COVID-19 era? Most importantly, would this form of missingness not be possibly quite problematic since data affected may actually be unknown to be missing (e.g., missing details of the medical history never recorded in the institution's EHR system)?

3. All patients who died within 24 hours of hospital admission were excluded. Can the authors report the number of such patients? Is antecedent statin use also known for these patients? It would seem that restricting the analysis to survivors could lead to differential exclusion of patients from the exposure groups based on post-exposure events, and that this may result in a biased comparison.

4. Were the treated patients used as reference, and untreated patients found to match exposed patients? This detail informs the interpretation of the estimated effect of treatment, since the

reference population over which an average treatment effect is obtained then consists of those patients who would have normally been on statins (and not simply patients satisfying the inclusion criteria for this study). Also, the use of a caliper matching approach means that some treated patients were excluded entirely, thereby further modifying the reference population with respect to which conclusions are made. The definition (and clear articulation) of the reference population is especially critical when it is believed that treatment may have a heterogeneous effect on different subpopulations of individuals.

5. My understanding is that the logistic regression model fitted on the matched subcohort is univariable – is this correct? This should be made explicit. If so, why is an odds ratio reported when it is as easy to report a relative risk or absolute risk? This could be done by taking a ratio or difference of subgroup-specific means if no covariate adjustment is needed. Would such measures not be much more interpretable? This is especially important given that the results of this study would likely be reported in mass media, where a ratio of odds is likely to be misinterpreted to be as a ratio of probabilities.

6. Because the interpretation of the treatment coefficient in the logistic regression model changes depending on whether or not other variables are included in the model, it is not particularly instructive to compare the results of analyses with and without adjustment since they are addressing different questions. In order to meaningfully compare the results from the entire cohort versus the matched subcohort, I would suggest either (i) also fitting the same multivariable logistic regression model used in the subcohort on data from the entire cohort, and comparing the treatment coefficient estimates from the two fits (subcohort only vs entire cohort); or (ii) using regression standardization to obtain treatment effect estimates that are interpretable and comparable across different models. See, for example, the commentary by Vansteelandt & Keiding (2011, *American Journal of Epidemiology*; 'G-Computation—Lost in Translation?') for a simple description of the idea. (Note: While in their commentary, these authors combine regression modelling with inverse-weighting using the propensity score, these ideas are equally applicable if inverse-weighting is replaced by matching.) Implementation of regression standardization can be easily coded from scratch, or alternatively, the `stdReg` package in R could be used. For another brief description of regression standardization and details about this package, see Sjolander (2016, *European Journal of Epidemiology*; 'Regression Standardization with the R package `stdReg`').

7. For analyses of duration data, it would seem that the interpretation of results would be significantly complicated by the fact that there is a competing risk of death. For example, it is easy to imagine that a treatment that effectively reduces disease mortality could make many patients who would otherwise have died early on remain hospitalized or intubated for longer periods of time, thereby increasing length of hospital stay or duration of invasive mechanical ventilation. Thus, appropriate context must be provided and caveats made when interpreting this type of analysis.

8. There are several mentions of hazard ratios in the paper, but it does not appear that any hazard-based model (e.g., Cox model) that would allow conclusions on the hazard scale was fitted here. This should be clarified.

9. The authors state that "Lipid levels were available for only 32% of the cohort. As such, we have presented them only at baseline." What are the implications of this massive amount of missingness for the use of lipid levels in the analyses? This is not clear to me from the text. Also, while it is indicated that BMI and insurance information were imputed, how was missing in other variables dealt with and why?

10. There are references to both 'gender' and 'sex' in the paper, which appear to be used interchangeably. However, these two terms have a different meaning. Which did the authors intend to use here?

Reviewer #1 (Remarks to the Author):

Comment 1:

Using data from Electronic Health Records, this study did a retrospective investigation to evaluate the effect of statins use on lowering the in-hospital mortality rate of COVID-19. Despite intensive discussions on this topic recently, real-world evidence demonstrating the effect of statins on COVID-19 is still lacking. This paper is of clinical importance, but can be improved by addressing the following comments.

1. There are a few recently published studies investigating the effect of statins use on COVID-19. Although the population might not be the same as this study, I think the authors should compare the results and provide some discussions. For example, Kow CS, Hasan SS. Meta-analysis of Effectiveness of Statins in Patients with Severe COVID-19. American Journal of Cardiology. 2020 Aug 12; Rodriguez-Nava, G., Trelles-Garcia, D.P., Yanez-Bello, M.A., Chung, C.W., Trelles-Garcia, V.P. and Friedman, H.J., 2020. Atorvastatin associated with decreased hazard for death in COVID-19 patients admitted to an ICU: a retrospective cohort study. Critical Care, 24(1), pp.1-2.

RESPONSE:

We thank the Reviewer for their thoughtful review and for this excellent suggestion. We have revised the discussion to now mention the referenced meta-analysis by Kow et al (which includes the second suggested study by Rodriguez-Nava and colleagues) as well as an ongoing randomized clinical trial in which we are evaluating the use of statins in COVID-19 patients:

“Most recently, a meta-analysis of 8,990 patients from 4 retrospective studies (including the study by Zhang and colleagues⁴⁶) revealed that COVID-19 patients who were statin users experienced significantly lower hazard for death or severe disease compared with non-statin users (hazard ratio 0.70, 95%CI 0.53-0.94)⁴⁷. Notably, these studies varied in terms of timing (inpatient vs. outpatient) as well as drug and dosing of statin regimens. Importantly, as in the study by Zhang and colleagues⁴⁶, in-hospital statin use in an observational setting may be subject to immortal time bias. With these studies as well as the findings of the present analysis in mind, the results of ongoing randomized clinical trials and registries will be crucial (Clinicaltrials.gov Identifiers: NCT04407273, NCT04390074, NCT04348695, NCT04426084, NCT04333407, NCT04380402). As such, many of the contributors to the current are participating in the undertaking of the INSPIRATION-S randomized clinical trial (NCT04486508)⁴⁸.”

Comment 2:

2. The secondary outcome is “a composite of in-hospital mortality or invasive mechanical ventilation within 30-days of admission”. So, my understanding is that the authors combined in-hospital mortality and invasive mechanical ventilation, which does not make sense here. Why not just use invasive mechanical ventilation within 30-days as the secondary outcome? From Table 3, it is clear that statins use is only associated with mortality but not the invasive mechanical ventilation, if we subtract the primary endpoints from the secondary endpoints. The significant association of the second endpoint comes from the association between the primary endpoint and statins use.

RESPONSE:

We appreciate the Reviewer's suggestion. We have now modified our secondary endpoint as follows – invasive mechanical ventilation at 30 days. As shown in Table 4, statin use tended to be associated with lower risk of invasive mechanical ventilation in our multivariable regression models in overall cohort (OR 0.80, 95% CI 0.64 – 1.02), and in our propensity-matched model (OR 0.82, 95% CI 0.62 – 1.07), but was not statistically significant.

Comment 3:

3. In Table 3, the p-value of oxygen saturation is a bit counterintuitive. The two groups have the same mean for oxygen saturation (both 94.0), why the p-value is so small? Compared with oxygen saturation, temperature has bigger mean difference and smaller confidence interval (which implies smaller variance), why the p-value is larger than oxygen saturation?

Response:

We thank the Reviewer for the attention to detail. On reviewing our oxygen saturation data, we identified some outlier values that were influencing the mean. We were able to cross check these outlier values against the EHR and fix them. The p- value after revising the data is 0.06 instead of 0.038.

Comment 4:

4. The data used in this analysis also provide opportunities for mediation analysis to understand how statins impact the mortality of COVID-19 patients. Since the lipid levels were observed, it would be interesting to investigate whether statins reduce the risk of mortality through lowering lipid level or other potential biological pathways.

Response:

We thank the Reviewer for this suggested analysis. While we certainly agree that a mechanistic understanding regarding how statin usage may contribute to better outcomes in patients with COVID-19 is necessary, this present analysis is limited by the fact that it is a retrospective evaluation of our institution's electronic medical record. As noted in the methods section on page 4, only 32% of patients had lipid profile data available, limiting our ability to perform mediation analyses. In addition, these lipid data were variable collected either prior to or during patient hospitalizations for COVID-19 and may or may not have been reflective of the initiation of statins prior to hospitalization. Therefore, we believe such an analysis would not be feasible using the current dataset.

Reviewer #2 (Remarks to the Author):

Gupta et al. performed a retrospective analysis of patients admitted with COVID-19 from February 1 to May 12 2020, to determine whether antecedent statin use was associated with lower 30-day in-hospital mortality in patients with confirmed COVID-19 infection. They report results on a total of 2,626 patients and a sub-cohort of 1,296 propensity-matched patients on 30-day hospital mortality (primary endpoint) and multiple secondary outcomes including a combination of hospital-mortality and various organ support measures. Based on their analysis, the authors demonstrate a statistically significant association of antecedent statin use with lower 30-day hospital mortality and advocate future prospective randomized controlled trials to confirm these findings.

This is an interesting and hypothesis generating study, particularly given the scarcity of treatment options and preventive therapeutic strategies for COVID-19. Statins are readily available and cheap, and these findings are encouraging and warrant further confirmation.

Comment 1:

1. Statins may affect various aspects of COVID-19 – reducing risk of acquiring COVID-19 infection, reducing severity and need for hospitalization, and once hospitalized, may reduce risk of poor outcomes. Although the authors identified statin use at home, it is not possible to draw inferences about statins reducing risk of acquiring COVID-19 infection or severity to prevent hospitalization since only hospitalized patients were assessed.

RESPONSE:

We thank the Reviewer for their careful review of the present manuscript. We completely agree that the findings of the current study should not be extrapolated to outpatients with regards to preventing COVID-19 infection or hospitalization, given that the analyses in this manuscript are restricted to hospitalized patients. Our findings and inferences only pertain to association of statin use with clinical outcomes in patients who are hospitalized with COVID-19. We have performed a thorough review of the manuscript and do not believe we have drawn any inferences, which would extend our findings to other patient populations. If there are specific areas which the Reviewer would suggest revising based on this suggestion, please let us know and we can adjust accordingly.

Comment 2:

2. There are several concerns with regards to the analyses to determine the association between statins and in-hospital mortality.

a) The focus should be on patients who received statins during the hospitalization rather than all patients who received statins at home.

RESPONSE:

While we appreciate the Reviewer's suggestion, performing such an analysis would be subject to immortal time bias as noted in our description of other studies referenced in the Discussion section. As detailed in this report by Lévesque and colleagues (BMJ 2010; 340; doi: <https://doi.org/10.1136/bmj.b5087>), immortal time bias is the circumstance in which determination of treatment status may involve a delay or waiting period when follow-up time is accrued. For example, patients had to have survived to the point at which they were dosed statins in the hospital, and this may introduce survival bias as these patients may have been healthier/less critically ill, thus inappropriately leading to favorable outcomes in the statin group. As such we chose to focus our study on outpatient statin use. However, to make sure our results remain consistent, we have now analyzed our primary endpoint using multivariable regression in the overall cohort with 'inpatient statin use' as the primary dependent variable of interest and included as our Supplemental Figure 5. As shown below, the association of inpatient statin use with in-hospital mortality at 30 days remains consistent.

Comment 3:

b) Statin users were older and had more chronic diseases. They may have presented earlier after symptom onset or more likely to be admitted – hence resulting in lead time bias. Non-statin users had higher point estimates for several important biomarkers (table 2 suggests they may be sicker at presentation), even after propensity matching, which corroborates this concern. Perhaps propensity matching of patients based on pre-admission characteristics and vital signs on presentation (Table 2). While this approach may include some variables that may mediate the beneficial effects of statins, it would address the issue of lead-time bias.

RESPONSE:

We understand the Reviewer’s concern regarding non-statin users being possibly sicker at presentation than statin users based on certain biomarkers and vital signs. However, one of the biomarkers that was lower in statin users was the C-reactive protein which mediates statin effect, like the Reviewer acknowledged, and as such we believe that it would not be methodologically sound to match on presentation characteristics. However, based on the reviewer’s suggestion, we did perform sensitivity analysis and matched our data on presenting vital signs and biomarkers. Our effect size for association of statin use with primary endpoint of in-hospital mortality at 30 days remains robust in the PS-matched model with OR 0.42 (0.32-0.55). We will defer to the Editors if they would like us to include this analysis in the main manuscript.

Comment 4:

c) Was the median time between COVID-19 testing (presumed symptom onset) and hospital admission similar in both groups. Were all COVID-19 tests performed in the emergency department or were patients referred to the ED after community-based testing?

RESPONSE:

Given that this database consists of patients that were admitted to a large, urban, academic medical center in the midst of the main COVID-19 surge in New York City and relatively early in the mass availability of RT-PCR testing, the majority of testing was performed while patients were in the confines of the hospital (either emergency room or in the newly established tents outside the hospital to aid triaging or as inpatient). The median time between time of specimen collection (presumed symptom onset as suggested by the Reviewer) and hospital admission was 8 hours in both groups (IQR 5 – 23 hours in statin users and IQR 5 to 17 hours in non-statin users).

Comment 5:

d) No information is provided on the quality of patient care other than organ support measures during hospitalization. Did statin and non-statin users receive similar supportive care and COVID-19 treatments? Therapeutic anticoagulation? Steroids? Fluid management? Or were statin-users perhaps 'more aggressively' treated early on based on their perceived higher risk?

RESPONSE:

We appreciate the Reviewer's concern regarding perceived higher risk in statin users and possible differences in care. While we do not believe that there were differences in quality of care in the 2 groups, we did look at rate of inpatient steroid use with the following results:

Statin Users 30.5% (290/951)
Non-Statin Users 32.0% (536/1675)
Chi squared p-value 0.45

The other treatments that the Reviewer mentioned like fluid management and therapeutic anticoagulation would require extensive data cleaning that would fall outside the scope of this study.

Comment 6:

e) Quality of outpatient and inpatient care may not only vary within hospitals, but also across hospitals. The authors should consider accounting for hospital or 'center' effect.

RESPONSE:

The Reviewer suggests looking at variation between hospitals. Our analysis includes 2 hospitals – The Columbia University Irving Medical Center (CUIMC) which is an academic teaching center and the Allen Hospital (AH) which is a community hospital, both of which are a part of the NewYork-Presbyterian Hospital system.

CUIMC – 1899 patients (694 statin users, 1205 statin non-users)
AH – 718 patients (256 statin users, 463 statin non-users)

We have now included a variable for ‘Community Hospital (AH)’ with reference as teaching hospital – CUIMC in our multivariable models for overall cohort for primary endpoint as shown in Figure 1 in manuscript. Our findings suggest that the hospital in which patients were treated were not associated with inpatient mortality at 30 days.

Community Hospital (AH) compared with Teaching Hospital (CUIMC):

Effect size for primary endpoint (in-hospital mortality at 30 days) – OR 1.03 (95% CI 0.8 – 1.3)

Comment 7:

3. Other concerns.

a) The time on ventilator and 30-day hospital mortality could be combined as an alternative secondary outcome of ‘ventilator-free days.’

RESPONSE:

We appreciate the Reviewer’s suggestion. Per Reviewer 1 Comment 2, we have now revised our secondary endpoint to ‘invasive mechanical ventilation at 30 days’. We will defer the preferred secondary endpoint to Editors and can change it if required.

Comment 8:

b) Discussion could be shortened.

RESPONSE:

Despite adding to the Discussion to meet the suggestions of Reviewer #1, we have shortened the Discussion section by >50 words in this current draft. At this point, we believe shortening this section further will limit our ability to speak to the use of statins in prior settings (including other ARDS presentations) and adequately discuss potential mechanisms for their benefit in patients with COVID-19.

Comment 9:

c) I didn’t see description of statin type or dose. Was a dose-response relationship observed?

Response:

We appreciate the Reviewer’s comment. While the information pertaining to type and dose of statins used would be highly insightful, examining a dose-response relationship is outside the scope of the current work. The purpose of this manuscript is to mainly assess the overall relationship between statin use and outcomes, we believe that a dose-dependent mechanistic paper is the next step and we are working to clean and process our data to be able to answer that question.

Reviewer #3 (Remarks to the Author):

This paper seeks to characterize the effect of statin use on COVID-19 health outcomes using data from a retrospective cohort assembled from the electronic health records of an academic center. Propensity score matching methods were used to account for the observational nature of this study.

The topic of this study is certainly salient and timely. There has been mounting evidence that statins may indeed be protective of various adverse outcomes in the context of COVID-19. This study could add one more pixel to the emerging picture, albeit one that may only be suggestive since it is based on observational data (and indications for statins are themselves strong risk factors for severe COVID-19). Ongoing clinical trials will hopefully provide a more definitive assessment and clinical guidance in due time.

There are several comments/questions that I noted while reading this manuscript.

Comment 1:

1. The definition of the exposure is a bit perplexing to me for several reasons.

a) Are patients deemed to be antecedent statin users if they have ever been recorded to use statins, or does the definition limit how far back in time such a record must have been made? It would help to further clarify the definition of an antecedent statin user.

Response:

We thank the Reviewer for this suggestion. As described in our Methods section, these were extracted from the medication reconciliation fields in the electronic medical record, which are entries that are updated at the time of hospital admission. As such, outpatient medications including statins used in our analysis are based on medication reconciliation done at the time of admission either with the patients or their families or their pharmacies. We cannot confirm duration of statin therapy prior to admission and have included this in our Limitation section.

Methods:

'Outpatient medications, including statins, angiotensin-converting enzyme inhibitors (ACEi), angiotensin receptor blockers (ARB), beta-blockers, oral anticoagulants, and P2Y12 inhibitors were extracted from medication reconciliation fields in the electronic medical record, which are entries that are updated at the time of hospital admission.'

Limitation:

'Moreover, it was not possible to verify duration of statin therapy or patient adherence with statin therapy. However, patients in the antecedent statin group had better lipid profiles, suggestive of medication effect.'

Comment 2:

b) It is unclear to me what potential intervention(s) involving statins the authors hypothesize as being possibly useful. For example, are they imagining an eventual recommendation for all at-risk individuals to be prescribed/administered statins as prophylaxis, at first symptoms of infection, or at hospital admission? This is important since it determines the relevant definition of the exposure, and statistical analyses should reflect this definition.

RESPONSE:

As the present analysis is a retrospective analysis of electronically collected medical record data, no formal recommendations can be made based on these findings and the manuscript is meant to be only hypothesis generating. However, the present analysis aims to assess whether prior statin use was associated with

any beneficial effect with regards to mortality or need for mechanical ventilation when admitted to our institution for treatment of COVID-19. As suggested in the Discussion section, several ongoing prospective RCTs are evaluating the use of statins, largely in cohorts of patients who are hospitalized with COVID-19, and this includes the prospective RCT INSPIRATION-S, an RCT involving statin use which the lead authors and several other investigators on this manuscript are contributors towards. However, we envision at least three clinical scenarios which require further clinical investigation to assess the potential utility of statin therapy in the setting of COVID-19: 1) prophylactic use in the setting of suspected or known exposure, 2) early COVID-19 infection in outpatients, and 3) patients with COVID-19 presenting/admitted to the hospital. We anticipate forthcoming data from RCTs to help provide prospectively collected data to help answer questions in these important clinical settings.

Comment 3:

c) In its current definition, the exposure appears to (possibly) occur over a (patient-specific) length of time prior to baseline rather than at baseline. In the propensity score model, variables that are in the causal pathway between statin use and COVID-19 outcomes should not be included. However, antecedent statin use not only precedes (and thus may affect) baseline patient characteristics, but also certain pre-baseline measurements obtained from the medical records. It would seem important to provide the hypothesized causal diagram underlying the mechanisms under study, and to comment on the adequateness of adjusting for each variable currently included in the model.

RESPONSE:

We agree that antecedent statin use was defined as prescription of statins prior to admission to our institution for COVID-19. However, we want to clarify that propensity adjustment was performed to account for the likelihood of antecedent statin administration (to control for confounding by indication of statin use). Therefore, we constructed a multivariable logistic regression model to predict the propensity of antecedent statin using selected clinical variables that may have influenced statin usage as covariates. We then implemented propensity matching to build a cohort matched on all these key variables between statin users and non-users. As shown in Supplemental Figure 2 included here, we were able to obtain a good covariate balance in our propensity-matched cohort.

As has been described in prior guidance statements for selecting variables in clinical trials or statistical analysis of clinical data, these variables were selected based on clinical rationale for association with primary endpoint and adverse events (European Medicines Agency. Guideline on adjustment for baseline covariates in clinical trials 2015).

Comment 4:

2. It is not clear to me what the source of medical records used to gather exposure and other patient data is. I presume records used are from the academic center in question. However, since it does not appear that this center is part of a closed, integrated care system (of the Kaiser-Permanente type), I wonder how complete such records are, and what population those patients for whom records are indeed available are representative of. Are most patients in this cohort seen in primary care at this same academic center? Additionally, for information obtained at admission, would there not be the risk of informative missingness, since patients admitted in more severe condition may be unable to provide much information, if any at all, particularly given restrictions on access to ERs and ICUs in the COVID-19 era? Most importantly, would this form of missingness not be possibly quite problematic since data affected may actually be unknown to be missing (e.g., missing details of the medical history never recorded in the institution's EHR system)?

RESPONSE:

The electronic medical record from the Columbia University Irving Medical Center and Allen Hospital, both academic medical center sites of the NewYork-Presbyterian Hospital system, were the data sources in the present analysis. As described in the Methods section, data was extracted using the institution's clinical data warehouse without any manual charge abstraction. Many of the patients who presented to our institution regularly follow in our primary care system, but certainly, there were many first-time patients who presented in the setting of the peak of the surge of this national/international health crisis due to the COVID-19 pandemic. While there is potential for missing data, we expect

medication reconciliation to have been undertaken whenever possible given the circumstances as part of best clinical practices. While there were visitor restrictions enacted at the peak of the pandemic, as many of our writing team members were on the front lines during this crisis, we are certain that clinical teams did their best to collect as much information as possible by contacting family/surrogates whenever feasible. These limitations regarding missing data are inherent to research using electronic medical record data. However, the use of electronic medical records for large-scale analyses is a well-established practice despite these limitations (Hemingway et al. European Heart Journal 2019; 39; <https://doi.org/10.1093/eurheartj/ehx487>), including several seminal COVID-19 publications both from our institution (Geleris et al. New England Journal of Medicine 2020; 382; doi: 10.1056/NEJMoa2012410) as well as other hospital systems (Reynolds et al. New England Journal of Medicine; 382; doi: 10.1056/NEJMoa2008975). With this being said, we believe that given the limited available prospectively collected data from the peak of the COVID-19 crisis in New York City, this present retrospective analysis from the electronic medical record is meaningful and has the potential to inform future studies in this space.

Comment 5:

3. All patients who died within 24 hours of hospital admission were excluded. Can the authors report the number of such patients? Is antecedent statin use also known for these patients? It would seem that restricting the analysis to survivors could lead to differential exclusion of patients from the exposure groups based on post-exposure events, and that this may result in a biased comparison.

Response:

As noted in our Methods section, 'patients who were admitted for less than 24 hours were excluded from this analysis.' This was done to ensure that our analysis is restricted to strictly patients who required hospital admission, as several patients could be discharged from the emergency department.

As such, among 1508 patients who were in the hospital for less than 24 hours, 267 patients died. We examined differences in death rates by statin use in this excluded population.

	Statin Users	Non-Statin Users
Death	60/397 (15.1%)	207/1111 (18.6%)

Chi-squared p-value = 0.13

As such, we find that death rate was not significantly different between statin users and non-statin users who were discharged within 24 hours and excluded from our analysis. For our results to be biased based on differential exclusion, death rate in excluded statin-users would need to have been significantly higher than that in excluded non-statin users. On the contrary, our findings show that absolute death rate was numerically lower in statin users, albeit statistically not significant. As such, these findings suggest that our results are not biased by differential exclusion.

Comment 6:

4. Were the treated patients used as reference, and untreated patients found to match exposed patients? This detail informs the interpretation of the estimated effect of treatment, since the reference population over which an average treatment effect is obtained then consists of those patients who would have normally been on statins (and not simply patients satisfying the inclusion criteria for this study). Also, the use of a caliper matching approach means that some treated patients were excluded entirely, thereby further modifying the reference population with respect to which conclusions are made. The definition (and clear articulation) of the reference population is especially critical when it is believed that treatment may have a heterogenous effect on different subpopulations of individuals.

Response:

The Reviewer's comment is not entirely clear to the authors and we would appreciate some clarification. In general, the reference population is non-statin users and the exposure variable is statin use. All comparisons in our manuscript are those of statin users to non-statin users. Our Methods section indicates that the study exposure variable is antecedent statin use.

Methods Section:

'Study exposure. The exposure in this study was antecedent statin use.'

The Reviewer also brings up exclusion of patients based on propensity matching. In general, when constructing propensity models and matching the population based on pre-specified calipers, some treated patients are dropped from the propensity-matched cohort. However, in order to make sure that our results are robust, we have also included multivariable regression models for the primary and secondary endpoints using the entire cohort as shown in Table 4 of our manuscript.

Table 4. Associations between Statin Use with Primary and Second Endpoints in Propensity-Matched Cohort and Multivariable-Adjusted Overall Cohorts of Patients Hospitalized with COVID-19

Primary endpoint – In-hospital mortality within 30 days		
	OR	95% CI
PS-matched	0.48	0.36 – 0.63
Multivariable (PS-matched)	0.46	0.34 – 0.62
Multivariable (overall)	0.49	0.38 – 0.63
Secondary endpoint – Invasive mechanical ventilation within 30 days		
	OR	95% CI
PS-matched	0.82	0.62 – 1.07
Multivariable (PS-matched)	0.80	0.61 – 1.06
Multivariable (overall)	0.80	0.64 – 1.02

CI = confidence interval, OR = odds ratio, PS = propensity scoring

Comment 7:

5. My understanding is that the logistic regression model fitted on the matched subcohort is univariable – is this correct? This should be made explicit. If so, why is an odds ratio reported when it is as easy to report a relative risk or absolute risk? This could be done by taking a ratio or difference of subgroup-specific means if no covariate adjustment is needed. Would such measures not be much more interpretable? This is especially important given that the results of this study would likely be reported in mass media, where a ratio of odds is likely to be misinterpreted to be as a ratio of probabilities.

Response:

The Reviewer is correct about the logistic regression model fitted on the propensity-matched cohort to be univariable. We chose to use logistic regression to remain consistent in our methods across all models. However, we also did report absolute risk in our Table 3. We have included it here as well.

Table 3. Clinical Outcomes in the Propensity-matched Cohort of Patients Hospitalized with COVID-19

	Statins Use (n = 648)	No Statins Use (n = 648)	P-Value
Primary endpoint	96 (14.8%)	172 (26.5%)	<0.001
Secondary endpoint	121 (18.6%)	142 (21.9%)	0.17
In-hospital mortality (anytime)	112 (17.2%)	201 (31.0%)	<0.001
Mechanical ventilation (anytime)	130 (20.1%)	158 (24.4%)	0.07
Vasopressor use	151 (23.3%)	200 (30.9%)	<0.01
CVVH	37 (5.7%)	45 (6.9%)	0.42
Length of hospital stay (days)	7.0 (4.0 – 12.0)	7.0 (3.0 – 14.0)	0.27
Days on ventilator	13.5 (3.8 – 31.6)	12.8 (2.6 – 34.7)	0.77

Data presented as N (%) or median (IQR).

Comment 8:

6. Because the interpretation of the treatment coefficient in the logistic regression model changes depending on whether or not other variables are included in the model, it is not particularly instructive to compare the results of analyses with and without adjustment since they are addressing different questions. In order to meaningfully compare the results from the entire cohort versus the matched subcohort, I would suggest either (i) also fitting the same multivariable logistic regression model used in the subcohort on data from the entire cohort, and comparing the treatment coefficient estimates from the two fits (subcohort only vs entire cohort); or (ii) using regression standardization to obtain treatment effect estimates that are interpretable and comparable across different models. See, for example, the commentary by Vansteelandt & Keiding (2011, American Journal of Epidemiology; ‘G-Computation—Lost in Translation?’) for a simple description of the idea. (Note: While in their commentary, these authors combine regression modelling with inverse-weighting using the propensity score, these ideas are equally applicable if inverse-weighting is replaced by matching.) Implementation of regression standardization can be easily coded from scratch, or alternatively, the stdReg package in R could be used. For another brief description of regression standardization and details about this package, see Sjolander (2016, European Journal of Epidemiology);

'Regression Standardization with the R package stdReg').

Response:

The Reviewer suggests that it may not be apt to compare point estimates from the multivariable adjusted logistic regression model with the univariable logistic regression of the propensity-matched cohort. We would like to clarify that our goal is not to compare estimates from these 2 strategies, but to make sure that our findings are robust and remain consistent across different statistical methodologies. As suggested by the reviewer, we have also provided the point estimate of statin use when fitting a multivariable regression model in the propensity-matched cohort, and our results remain robust. The point estimates do not change much with multivariable adjustment in PS-matched cohort as we obtained a good covariate balance in propensity adjustment, which accounts for differences in these variables.

Table 4. Associations between Statin Use with Primary and Second Endpoints in Propensity-Matched Cohort and Multivariable-Adjusted Overall Cohorts of Patients Hospitalized with COVID-19

Primary endpoint – In-hospital mortality within 30 days		
	OR	95% CI
PS-matched	0.48	0.36 – 0.63
Multivariable (PS-matched)	0.46	0.34 – 0.62
Multivariable (overall)	0.49	0.38 – 0.63

Secondary endpoint – Invasive mechanical ventilation within 30 days		
	OR	95% CI
PS-matched	0.82	0.62 – 1.07
Multivariable (PS-matched)	0.80	0.61 – 1.06
Multivariable (overall)	0.80	0.64 – 1.02

CI = confidence interval, OR = odds ratio, PS = propensity scoring

Comment 9:

7. For analyses of duration data, it would seem that the interpretation of results would be significantly complicated by the fact that there is a competing risk of death. For example, it is easy to imagine that a treatment that effectively reduces disease mortality could make many patients who would otherwise have died early on remain hospitalized or intubated for longer periods of time, thereby increasing length of hospital stay or duration of invasive mechanical ventilation. Thus, appropriate context must be provided and caveats made when interpreting this type of analysis.

Response:

Thank you for this suggestion. We have added the following text to the Limitations section “While the primary endpoint of in-hospital mortality was significantly lower in antecedent statin users, it remains to be seen whether patients who

survived (possibly in-part due to prior statin therapy) may experience long-term morbidity and sequelae of COVID-19 infection, and further analyses are needed in this regard.”

Additionally, it is important to note that there was no significant difference in median length of stay between antecedent statin users and non-statin users in the propensity matched model (7.0 [4.0-12.0] vs. 7.0 [3.0-14.0], p=0.27) as noted in Table 3.

Comment 10:

8. There are several mentions of hazard ratios in the paper, but it does not appear that any hazard-based model (e.g., Cox model) that would allow conclusions on the hazard scale was fitted here. This should be clarified.

RESPONSE:

We thank the Reviewer for making this distinction. Earlier versions of the analysis used Cox proportional hazards modeling; however, as the reviewer noted, the current manuscript uses logistic regression modeling. We have now corrected the manuscript to read as odds ratios throughout the text.

Comment 11:

9. The authors state that “Lipid levels were available for only 32% of the cohort. As such, we have presented them only at baseline.” What are the implications of this massive amount of missingness for the use of lipid levels in the analyses? This is not clear to me from the text. Also, while it is indicated that BMI and insurance information were imputed, how was missing in other variables dealt with and why?

Response:

We appreciate the Reviewer’s comment. Lipid levels were not usually checked in patients hospitalized with COVID-19, which explains why a large proportion of these are not available. However, lipid levels were not crucial to our analysis, as we did not perform a mediation analysis. Our objective was to examine the association of antecedent statin use with clinical outcomes, and do not need lipid levels to do so. As described in our section on ‘Missing Data’, variables other than race, BMI and insurance were missing for less than 5% of the population and did not require imputation.

Methods section:

‘Missing data. BMI and insurance information were missing in 19% and 15% of the patients, respectively, and multiple imputation with predictive mean matching was utilized to adjust the models for BMI and insurance. We imputed one hundred datasets, fitted the logistic regression models for the primary and secondary endpoints for each imputed dataset, estimated the odds ratios on each imputed dataset, and then averaged the one hundred estimated values to obtain the pooled estimates. Model estimates and standard errors were calculated with Rubin’s rules¹¹. Race and ethnicity were missing in 30% of the patients and were classified as ‘others/missing’ while adjusting in the models. Lipid levels were available for only 32% of the cohort. As such, we have presented them only at baseline. The remaining variables were missing in fewer than 5% of the study cohort.’

Comment 12:

10. There are references to both 'gender' and 'sex' in the paper, which appear to be used interchangeably. However, these two terms have a different meaning. Which did the authors intend to use here?

RESPONSE:

We thank the Reviewer for making this important distinction. We have changed the one instance where gender was referenced to sex, as this is what we intended to use throughout and is reflective of what is collected from the electronic medical record.

Reviewers' Comments:

Reviewer #1:

Remarks to the Author:

The authors have addressed my previous comments and I do not have further comments.

Reviewer #2:

None

Reviewer #3:

Remarks to the Author:

Please see attached report.

MY COMMENTS/QUESTIONS ARE INTERSPERSED BELOW AND HIGHLIGHTED IN RED.

Reviewer #3 (Remarks to the Author):

Comment 1:

1. The definition of the exposure is a bit perplexing to me for several reasons.

a) Are patients deemed to be antecedent statin users if they have ever been recorded to use statins, or does the definition limit how far back in time such a record must have been made? It would help to further clarify the definition of an antecedent statin user.

Response:

We thank the Reviewer for this suggestion. As described in our Methods section, these were extracted from the medication reconciliation fields in the electronic medical record, which are entries that are updated at the time of hospital admission. As such, outpatient medications including statins used in our analysis are based on medication reconciliation done at the time of admission either with the patients or their families or their pharmacies. We cannot confirm duration of statin therapy prior to admission and have included this in our Limitation section.

Methods:

'Outpatient medications, including statins, angiotensin-converting enzyme inhibitors (ACEi), angiotensin receptor blockers (ARB), beta-blockers, oral anticoagulants, and P2Y12 inhibitors were extracted from medication reconciliation fields in the electronic medical record, which are entries that are updated at the time of hospital admission.'

Limitation:

'Moreover, it was not possible to verify duration of statin therapy or patient adherence with statin therapy. However, patients in the antecedent statin group had better lipid profiles, suggestive of medication effect.'

Does this imply that only a current prescription for statin use is considered in defining antecedent use upon admission? The verbiage is not entirely clear to me in this regard. If so, perhaps it would be clearer to say "Current prescriptions of outpatient medications, including..." to emphasize this?

Comment 4:

2. It is not clear to me what the source of medical records used to gather exposure and other patient data is. I presume records used are from the academic center in question. However, since it does not appear that this center is part of a closed, integrated care system (of the Kaiser-Permanente type), I wonder how complete such records are, and what population those patients for whom records are indeed available are representative of. Are most patients in this cohort seen in primary care at this same academic center? Additionally, for information obtained at admission, would there not be the risk of informative missingness, since patients admitted in more severe condition may be unable to provide much information, if any at all, particularly given restrictions on access to ERs and ICUs in the COVID-19 era? Most importantly, would this form of missingness not be possibly quite problematic since data affected may actually be unknown to be missing (e.g., missing details of the medical history never recorded in the institution's EHR system)?

RESPONSE:

The electronic medical record from the Columbia University Irving Medical Center and Allen Hospital, both academic medical center sites of the NewYork-Presbyterian Hospital system, were the data sources in the present analysis. As described in the Methods section, data was extracted using the institution's clinical data warehouse without any manual charge abstraction. Many of the patients who presented to our institution regularly follow in our primary care system, but certainly, there were many first-time patients who presented in the setting of the peak of the surge of this national/international health crisis due to the COVID-19 pandemic. While there is potential for missing data, we expect medication reconciliation to have been undertaken whenever possible given the circumstances as part of best clinical practices. While there were visitor restrictions enacted at the peak of the pandemic, as many of our writing team members were on the front lines during this crisis, we are certain that clinical teams did their best to collect as much information as possible by contacting family/surrogates whenever feasible. These limitations regarding missing data are inherent to research using electronic medical record data. However, the use of electronic medical records for large-scale analyses is a well-established practice despite these limitations (Hemingway et al. European Heart Journal 2019; 39;<https://doi.org/10.1093/eurheartj/ehx487>), including

several seminal COVID-19 publications both from our institution (Geleris et al. New England Journal of Medicine 2020; 382; doi: 10.1056/NEJMoa2012410) as well as other hospital systems (Reynolds et al. New England Journal of Medicine; 382; doi: 10.1056/NEJMoa2008975). With this being said, we believe that given the limited available prospectively collected data from the peak of the COVID-19 crisis in New York City, this present retrospective analysis from the electronic medical record is meaningful and has the potential to inform future studies in this space.

For the sake of transparency, it would seem important to at least acknowledge the potential for informative data missingness in the limitations of this study.

Comment 5:

3. All patients who died within 24 hours of hospital admission were excluded. Can the authors report the number of such patients? Is antecedent statin use also known for these patients? It would seem that restricting the analysis to survivors could lead to differential exclusion of patients from the exposure groups based on post-exposure events, and that this may result in a biased comparison.

Response:

As noted in our Methods section, ‘patients who were admitted for less than 24 hours were excluded from this analysis.’ This was done to ensure that our analysis is restricted to strictly patients who required hospital admission, as several patients could be discharged from the emergency department.

As such, among 1508 patients who were in the hospital for less than 24 hours, 267 patients died. We examined differences in death rates by statin use in this excluded population.

	Statin Users	Non-Statin Users
Death	60/397 (15.1%)	207/1111 (18.6%)

Chi-squared p-value = 0.13

As such, we find that death rate was not significantly different between statin users and non-statin users who were discharged within 24 hours and excluded from our analysis. For our results to be biased based on differential exclusion, death rate in excluded statin-users would need to have been significantly higher than that in excluded non-statin users. On the contrary, our findings show that absolute death rate was numerically lower in statin users, albeit statistically not significant. As such, these findings suggest that our results are not biased by differential exclusion.

It may be worth including a sentence or two in the paper to explain this.

Comment 6:

4. Were the treated patients used as reference, and untreated patients found to match exposed patients? This detail informs the interpretation of the estimated effect of treatment, since the reference population over which an average treatment effect is obtained then consists of those patients who would have normally been on statins (and not simply patients satisfying the inclusion criteria for this study). Also, the use of a caliper matching approach means that some treated patients were excluded entirely, thereby further modifying the reference population with respect to which conclusions are made. The definition (and clear articulation) of the reference population is especially critical when it is believed that treatment may have a heterogenous effect on different subpopulations of individuals.

Response:

The Reviewer’s comment is not entirely clear to the authors and we would appreciate some clarification. In general, the reference population is non-statin users and the exposure variable is statin use. All comparisons in our manuscript are those of statin users to non-statin users. Our Methods section indicates that the study exposure variable is antecedent statin use.

Below, I will try to clarify what my point was.

When estimating an average causal effect (or related measure), it is important to be explicit about the reference population over which the ‘average’ is taken. Since the causal effect will typically be different in differing subpopulations of patients, this reference population matters. For example, if the reference

population is taken to be the treated patients, then you are estimating an average effect among patients with similar characteristics to those who in your observational context had antecedent statin use upon admission. These patients are certainly different from patients who did not have antecedent statin use at admission, as you have shown in your study. If the reference population is taken to be the treated, your association/effect parameter is getting at the difference in mean outcomes that you would see if you were able to go back in time and intervene so that these patients did not have antecedent statin use at admission. This is different, for example, from imagining what would happen if you were able to assign all study patients to antecedent statin use versus to non-use, in which case the reference population would be the population of which all study patients are representative.

If your matching procedure seeks, for each treated patient, to find an appropriate match within the sample of untreated patients, the reference population is the treated patients. (This is what most software for matching actually do by default.) If instead your matching procedure seeks, for each untreated patient, to find an appropriate match within the treated patients, the reference population is the untreated patients. If the matching procedure instead seeks to find a match for each treated and untreated patient among untreated and treated patients, respectively, then the reference population is the entire study sample (though in this case there are other complicating issues in inference, e.g., validity of inference in the face of replicated observations).

At the end of the Methods section, you mention that you used the MatchIt package in R for matching. One of the arguments of the `matchit()` function is 'estimand'. If you did not specify a value for this argument when running the code for your analyses, then your reference population is necessarily the treated patients, since the default for this argument is the ATT (average treatment effect among the treated) as opposed to other alternatives. You mention in your response that the reference population you intended is the non-statin users admitted for COVID. Given how different these patients are from the statin users, using the treated as reference could result in a biased answer to the scientific question of interest. How off the result would be depends on the extent to which the effect of statins on COVID outcomes differs in statin users versus non-users. To get at the effect among non-users, you would need to redo the matching procedure with argument 'estimand=ATC' if indeed the original analyses were done with the default value of 'estimand' (here, ATC is an acronym that refers to average treatment effect among the controls). I leave it up to you (and the editors) whether or not it is worth rerunning the analyses in this manner. However, regardless of whether you do, there should at the very least be a statement in the Methods section indicating what is the reference population for the results of your matching-based analyses.

ADDITIONAL COMMENTS:

1. Since this is a fast-moving subject area, the literature review should be as up to date as possible before publication. There are additional studies of statins in COVID (published in the last few months) that should be appropriately referenced. I encourage you to do a quick search online to identify these.

2. (regarding Comment 1 of Reviewer 1 and Comment 2 of Reviewer 2)

You highlight immortal time bias as a motivation for not formally considering in-hospital statin use as an exposure of potential interest, and indeed, you are correct that this is a potential pitfall to consider. However, immortal time bias only occurs if an improper analytic approach is taken. Immortal time bias would be present, for example, if patients are (incorrectly) categorized as ever versus never initiating statin use while in the hospital; this is problematic since it means comparison groups are defined based upon post-baseline events. If instead the exposure (in-hospital initiation of statin use) is accounted for as a time-varying covariate (zero before initiation, one afterward), then immortal time bias would not arise, and it would be possible to assess the Reviewers' question.

3. (regarding Comment 2 of Reviewer 1)

I appreciate that it may be clinically interesting to discern between the effect of statins on time until mechanical ventilation and on time until death separately. Nevertheless, inferences based on the composite time are likely to be less fraught with biases. If time until mechanical ventilation is taken as secondary outcome of interest, then death would presumably be treated as a censoring event,

even though it is a highly informative competing risk. As such, the independent censoring assumption needed by the various statistical methods available are much more likely to be violated with such an outcome. I support the suggestion of Reviewer 2 (Comment 7) to use as secondary endpoint 'ventilator-free survival'.

4. In the Discussion section, the authors state that "as a retrospective analysis of electronic medical record data, there remains the potential for unmeasured confounders. However, we performed propensity matched analysis and multivariable adjustment to minimize the likelihood for confounding." It is important to note though that neither propensity matched analyses nor multivariable adjustment in any way reduce the risk of bias due to unmeasured confounders, for which there is essentially no remedy (apart from getting more data!).

Response to Reviewers

Reviewer #3 (Remarks to the Author):

Comment 1:

1. The definition of the exposure is a bit perplexing to me for several reasons.

a) Are patients deemed to be antecedent statin users if they have ever been recorded to use statins, or does the definition limit how far back in time such a record must have been made? It would help to further clarify the definition of an antecedent statin user.

Response:

We thank the Reviewer for this suggestion. As described in our Methods section, these were extracted from the medication reconciliation fields in the electronic medical record, which are entries that are updated at the time of hospital admission. As such, outpatient medications including statins used in our analysis are based on medication reconciliation done at the time of admission either with the patients or their families or their pharmacies. We cannot confirm duration of statin therapy prior to admission and have included this in our Limitation section.

Methods:

'Outpatient medications, including statins, angiotensin-converting enzyme inhibitors (ACEi), angiotensin receptor blockers (ARB), beta-blockers, oral anticoagulants, and P2Y12 inhibitors were extracted from medication reconciliation fields in the electronic medical record, which are entries that are updated at the time of hospital admission.'

Limitation:

'Moreover, it was not possible to verify duration of statin therapy or patient adherence with statin therapy. However, patients in the antecedent statin group had better lipid profiles, suggestive of medication effect.'

Does this imply that only a current prescription for statin use is considered in defining antecedent use upon admission? The verbiage is not entirely clear to me in this regard. If so, perhaps it would be clearer to say "Current prescriptions of outpatient medications, including..." to emphasize this?

Response:

We appreciate the Reviewer suggestion, and have now updated the verbiage as follows:

Methods:

"Outpatient medications, including statins, angiotensin-converting enzyme inhibitors (ACEi), angiotensin receptor blockers (ARB), beta-blockers, oral anticoagulants, and P2Y12 inhibitors were extracted from medication reconciliation fields in the electronic medical record, which are entries of current prescriptions that are updated at the time of hospital admission."

"Study exposure. The exposure in this study was antecedent statin use. Antecedent statin use was defined as record of current prescription of statins as a home medication in the electronic medical record. Home medications are typically reconciled with patients or their families or pharmacies at the time of admission."

Comment 4:

2. It is not clear to me what the source of medical records used to gather exposure and other patient data is. I presume records used are from the academic center in question. However, since it does not appear that this center is part of a closed, integrated care system (of the Kaiser-Permanente type), I wonder how complete such records are, and what population those patients for whom records are indeed available are representative of. Are most patients in this cohort seen in primary care at this same academic center? Additionally, for information obtained at admission, would there not be the risk of informative missingness, since patients admitted in more severe condition may be unable to provide much information, if any at all, particularly given restrictions on access to ERs and ICUs in the COVID-19 era? Most importantly, would this form of missingness not be possibly quite problematic since data affected may actually be unknown to be missing (e.g., missing details of the medical history never recorded in the institution's EHR system)?

RESPONSE:

The electronic medical record from the Columbia University Irving Medical Center and Allen Hospital, both academic medical center sites of the NewYork-Presbyterian Hospital system, were the data sources in the present analysis. As described in the Methods section, data was extracted using the institution’s clinical data warehouse without any manual chart abstraction. Many of the patients who presented to our institution regularly follow in our primary care system, but certainly, there were many first-time patients who presented in the setting of the peak of the surge of this national/international health crisis due to the COVID-19 pandemic. While there is potential for missing data, we expect medication reconciliation to have been undertaken whenever possible given the circumstances as part of best clinical practices. While there were visitor restrictions enacted at the peak of the pandemic, as many of our writing team members were on the front lines during this crisis, we are certain that clinical teams did their best to collect as much information as possible by contacting family/surrogates whenever feasible. These limitations regarding missing data are inherent to research using electronic medical record data. However, the use of electronic medical records for large-scale analyses is a well-established practice despite these limitations (Hemingway et al. European Heart Journal 2019; 39; <https://doi.org/10.1093/eurhearti/ehx487>), including several seminal COVID-19 publications both from our institution (Geleris et al. New England Journal of Medicine 2020; 382; doi: 10.1056/NEJMoa2012410) as well as other hospital systems (Reynolds et al. New England Journal of Medicine; 382; doi: 10.1056/NEJMoa2008975). With this being said, we believe that given the limited available prospectively collected data from the peak of the COVID-19 crisis in New York City, this present retrospective analysis from the electronic medical record is meaningful and has the potential to inform future studies in this space.

For the sake of transparency, it would seem important to at least acknowledge the potential for informative data missingness in the limitations of this study.

Response:

We appreciate this suggestion and have now added the following phrase to the Limitations section:

“In addition, missingness for disease and drug variables cannot be quantified, as there were no codes to indicate that data were missing. It was assumed that a characteristic was not present if the patient’s record did not include information on it, such as hypertension or the use of statins.”

Comment 5:

3. All patients who died within 24 hours of hospital admission were excluded. Can the authors report the number of such patients? Is antecedent statin use also known for these patients? It would seem that restricting the analysis to survivors could lead to differential exclusion of patients from the exposure groups based on post-exposure events, and that this may result in a biased comparison.

Response:

As noted in our Methods section, ‘patients who were admitted for less than 24 hours were excluded from this analysis.’ This was done to ensure that our analysis is restricted to strictly patients who required hospital admission, as several patients could be discharged from the emergency department.

As such, among 1508 patients who were in the hospital for less than 24 hours, 267 patients died. We examined differences in death rates by statin use in this excluded population.

	Statin Users	Non-Statin Users
Death	60/397 (15.1%)	207/1111 (18.6%)

Chi-squared p-value = 0.13

As such, we find that death rate was not significantly different between statin users and non-statin users who were discharged within 24 hours and excluded from our analysis. For our results to be biased based on differential exclusion, death rate in excluded statin-users would need to have been significantly higher than that in excluded non-statin users. On the contrary, our findings show that absolute death rate was numerically lower in statin users, albeit statistically not significant. As such, these findings suggest that our results are not biased by differential exclusion.

It may be worth including a sentence or two in the paper to explain this.

Response:

We appreciate the suggestion, and have now added a sentence within our Methods section and added the Table to the Supplemental document.

“The mortality rate was not significantly different between statin users and non-statin users who were discharged within 24 hours and excluded from our analysis (Supplemental Table 1).”

Comment 6:

4. Were the treated patients used as reference, and untreated patients found to match exposed patients? This detail informs the interpretation of the estimated effect of treatment, since the reference population over which an average treatment effect is obtained then consists of those patients who would have normally been on statins (and not simply patients satisfying the inclusion criteria for this study). Also, the use of a caliper matching approach means that some treated patients were excluded entirely, thereby further modifying the reference population with respect to which conclusions are made. The definition (and clear articulation) of the reference population is especially critical when it is believed that treatment may have a heterogeneous effect on different subpopulations of individuals.

Response:

The Reviewer’s comment is not entirely clear to the authors and we would appreciate some clarification. In general, the reference population is non-statin users and the exposure variable is statin use. All comparisons in our manuscript are those of statin users to non-statin users. Our Methods section indicates that the study exposure variable is antecedent statin use.

Below, I will try to clarify what my point was.

When estimating an average causal effect (or related measure), it is important to be explicit about the reference population over which the ‘average’ is taken. Since the causal effect will typically be different in differing subpopulations of patients, this reference population matters. For example, if the reference population is taken to be the treated patients, then you are estimating an average effect among patients with similar characteristics to those who in your observational context had antecedent statin use upon admission. These patients are certainly different from patients who did not have antecedent statin use at admission, as you have shown in your study. If the reference population is taken to be the treated, your association/effect parameter is getting at the difference in mean outcomes that you would see if you were able to go back in time and intervene so that these patients did not have antecedent statin use at admission. This is different, for example, from imagining what would happen if you were able to assign all study patients to antecedent statin use versus to non-use, in which case the reference population would be the population of which all study patients are representative.

If your matching procedure seeks, for each treated patient, to find an appropriate match within the sample of untreated patients, the reference population is the treated patients. (This is what most software for matching actually do by default.) If instead your matching procedure seeks, for each untreated patient, to find an appropriate match within the treated patients, the reference population is the untreated patients. If the matching procedure instead seeks to find a match for each treated and untreated patient among untreated and treated patients, respectively, then the reference population is the entire study sample (though in this case there are other complicating issues in inference, e.g., validity of inference in the face of replicated observations).

At the end of the Methods section, you mention that you used the MatchIt package in R for matching. One of the arguments of the matchit() function is ‘estimand’. If you did not specify a value for this argument when running the code for your analyses, then your reference population is necessarily the treated patients, since the default for this argument is the ATT (average treatment effect among the treated) as opposed to other alternatives. You mention in your response that the reference population you intended is the non-statin users admitted for COVID. Given how different these patients are from the statin users, using the treated as reference could result in a biased answer to the scientific question of interest. How off the result would be depends on the extent to which the effect of statins on COVID outcomes differs in statin users versus non-users. To get at the effect among non-users, you would need to redo the matching procedure with argument ‘estimand=ATC’ if indeed the original analyses were done with the default value of ‘estimand’ (here, ATC is an acronym that refers to average treatment effect among the controls). I leave it up to you (and the editors) whether or not it is worth rerunning the analyses in this manner. However, regardless of whether you do, there should at the very least be a statement in the Methods section

indicating what is the reference population for the results of your matching-based analyses.

Response:

We are grateful to the Reviewer for their detailed suggestions. We re-ran our analyses using the argument 'estimand=ATC' so that the control group is the reference for our analysis.

We have clarified in our Methods – “For the primary and secondary endpoints, we performed logistic regression on the propensity-matched cohort with the control group as reference.”

The results remained similar.

“The primary endpoint occurred in 96 (14.8%) patients receiving statins compared to 172 (26.5%) not receiving statins, (OR 0.47, 95% CI 0.36 – 0.62, p<0.001). The secondary endpoint occurred in 121 (18.6%) patients receiving statins compared to 142 (21.9%) not receiving statins, (OR 0.76, 95% CI 0.58 – 1.00).”

ADDITIONAL COMMENTS:

1. Since this is a fast-moving subject area, the literature review should be as up to date as possible before publication. There are additional studies of statins in COVID (published in the last few months) that should be appropriately referenced. I encourage you to do a quick search online to identify these.

Response:

We thank the reviewer for this suggestion. Indeed a few additional analyses have been completed, including an additional meta-analysis focused specifically on European and Western populations. Thus, the protective effect of statins (as determined by retrospective study) appears to be apparent in both predominantly Asian and Western patient populations. We have referenced this manuscript and updated the paragraph on other studies focusing on studies evaluating statin use in COVID-19 patients as follows:

“The limited evidence available regarding statins in the COVID-19 literature confirms the findings presented in the present manuscript. In a study which evaluated the prevalence and impact of myocardial injury in 2,736 hospitalized patients in New York City, 36% of patients received statins prior to admission (Paranjpe, I., et al. JACC 2020). Though not the focus of this manuscript, statin use was associated with significantly lower rates of in-hospital mortality by multivariable analysis (OR 0.57, 95% CI 0.47-0.69) (Paranjpe, I., et al. JACC 2020). Additionally, a separate study of 154 elderly individuals suggested that statin use prior to admission was associated with less severe symptoms, but they did not assess in-hospital mortality (De Spiegeleer A., et al. J Am Med Dir Assoc 2020). More recently, an analysis from the Wuhan, China demonstrated significantly lower 28-day mortality in patients who received inpatient statins compared with non-statin users (adjusted hazard ratio 0.58, 95%CI 0.43-0.80) (Zhang, X.J., et al. Cell Metab 2020). In this study, however, <10% of hospitalized patients received statins. A meta-analysis of 8,990 patients from 4 retrospective studies (including the study by Zhang and colleagues (Zhang, X.J., et al. Cell Metab 2020)) revealed that COVID-19 patients who were statin users experienced significantly lower hazard for death or severe disease compared with non-statin users (hazard ratio 0.70, 95%CI 0.53-0.94) (Kow, C.S., et al. Am J Cardiol 2020). As the majority of these studies focused on patients from China, they may not be representative of the patient characteristics and burden of cardiovascular comorbidities in Western populations. Most recently, a separate meta-analysis focused exclusively on European and North American patient populations, and only one of the seven studies included was common to the previously mentioned analysis by Kow and colleagues (Kow, C.S., et al. Am J Cardiol 2020 and Onorato, D., et al. Semin Thromb Hemos 2020). Statin use was associated with significantly lower rates of progression to severe COVID-19 illness or death (OR 0.59, 95%CI 0.35-0.99) (Onorato, D., et al. Semin Thromb Hemos 2020). Notably, studies included in both of these meta-analyses varied significantly in terms of patient populations, adjunctive therapies administered, timing of administration (inpatient vs. outpatient) as well as drug and dosing of statin regimens. Notably, studies included in both of these meta-analyses varied in terms of timing of patient populations, adjunctive therapies administered, timing of administration (inpatient vs. outpatient) as well as drug and dosing of statin regimens. Importantly, as in the study by Zhang and colleagues (Zhang, X.J., et al. Cell Metab 2020), in-hospital statin use in an observational setting may be subject to immortal time bias. With these studies as well as the findings of the present analysis in mind, the results of ongoing randomized clinical trials and registries will be

crucial (Clinicaltrials.gov Identifiers: NCT04407273, NCT04390074, NCT04348695, NCT04426084, NCT04333407, NCT04380402). As such, many of the contributors to the current report are participating in the undertaking of the INSPIRATION-S randomized clinical trial (NCT04486508) (Bikdeli, B. et al. Thromb Res 2020)."

1. (regarding Comment 1 of Reviewer 1 and Comment 2 of Reviewer 2)

You highlight immortal time bias as a motivation for not formally considering in-hospital statin use as an exposure of potential interest, and indeed, you are correct that this is a potential pitfall to consider. However, immortal time bias only occurs if an improper analytic approach is taken.

Immortal time bias would be present, for example, if patients are (incorrectly) categorized as ever versus never initiating statin use while in the hospital; this is problematic since it means comparison groups are defined based upon post-baseline events. If instead the exposure (in-hospital initiation of statin use) is accounted for as a time-varying covariate (zero before initiation, one afterward), then immortal time bias would not arise, and it would be possible to assess the Reviewers' question.

Response:

We appreciate the Reviewers suggestions. While we did not formally examine inpatient statin use, we did perform sensitivity analysis using inpatient statin use and the overall results are similar, as shown in Supplemental Table 5. Moreover, majority of patients who received antecedent statins also received inpatient statins as shown in Table 1. As such, we do not think that changing our entire analysis would be the most optimal approach at this time, given that the results are similar by both analytic strategies.

2. (regarding Comment 2 of Reviewer 1)

I appreciate that it may be clinically interesting to discern between the effect of statins on time until mechanical ventilation and on time until death separately. Nevertheless, inferences based on the composite time are likely to be less fraught with biases. If time until mechanical ventilation is taken as secondary outcome of interest, then death would presumably be treated as a censoring event, even though it is a highly informative competing risk. As such, the independent censoring assumption needed by the various statistical methods available are much more likely to be violated with such an outcome. I support the suggestion of Reviewer 2 (Comment 7) to use as secondary endpoint 'ventilator-free survival'.

Response:

We will defer to the Editors for this comment. Our initial submission included the secondary endpoint of 'combined in-hospital mortality and invasive mechanical ventilation at 30 days.' Based on comments from Reviewer 1, we modified our secondary endpoint to 'invasive mechanical ventilation at 30 days.' We believe that changing our secondary endpoint one more time based on this comment to 'ventilator-free survival' will revert the endpoint to a similar one included in the original submission.

3. In the Discussion section, the authors state that "as a retrospective analysis of electronic medical record data, there remains the potential for unmeasured confounders. However, we performed propensity matched analysis and multivariable adjustment to minimize the likelihood for confounding." It is important to note though that neither propensity matched analyses nor multivariable adjustment in any way reduce the risk of bias due to unmeasured confounders, for which there is essentially no remedy (apart from getting more data!).

Response:

We have now rephrased this as follows:

"We performed propensity matched analysis and multivariable adjustment to minimize the likelihood for confounding. As a retrospective analysis of electronic medical record data, however, there remains the potential for unmeasured confounders."